# Connected subglacial lake drainage beneath Thwaites Glacier, West Antarctica

Benjamin E. Smith[1], Noel Gourmelen[2], Alexander Huth[3], Ian Joughin[1]

[1]Applied Physics Lab, University of Washington, Seattle, WA, 98195, USA
[2]School of Geosciences, University of Edinburgh, Edinburgh, EH8, Scotland
[3]Department of Earth and Space Sciences, University of Washington, Seattle, WA, 98195, USA

*Correspondence to*: Benjamin Smith (bsmith@apl.washington.edu)

**Abstract.**  We present conventional and swath altimetry data from CryoSat-2, revealing a system of subglacial lakes that drained between June 2013 and January 2014 under the central part of Thwaites Glacier, West Antarctica (TWG).  Much of the drainage happened in less than six months, with an apparent connection between three lakes spanning more than 130 km.  Hydropotential analysis of the glacier bed shows a large number of small closed basins that should trap water produced by subglacial

melt, although the observed large-scale motion of water suggests that water can sometimes locally move against the apparent potential gradient, at least during lake-drainage events.  This shows that there are important limitations in the ability of hydropotential maps to predict subglacial water flow.  An interpretation based on a map of the melt rate suggests that lake drainages of this type should take place every 20-80 years, depending on the connectivity of the water flow at the bed.  Although we observed

an acceleration in the downstream part of TWG immediately before the start of the lake drainage, there is no clear connection between the drainage and any speed change of the glacier.

## 1. Background

The Amundsen Sea embayment is one of the fastest-changing part of Antarctica, with large changes since at least the 1990s (Rignot, 2008).  Increased flow of Thwaites Glacier (TWG) is

responsible for around half of the ice-sheet mass loss from this sector (Medley et al., 2014); in response to these large changes, NSF's AGASEA and NASA's IceBridge programs have flown extensive surveys measuring ice thickness and bed elevation in this area, with the twin goals of measuring mass-balance changes and enabling accurate ice-flow modelling for the region.  As a result, the bed of TWG has been mapped in detail, allowing mapping of basal shear stress and potential subglacial water flow

paths.  These reveal abundant basal meltwater production, estimated at about 3.5 $km^3yr^{-1}$ and averaging ~19 mm $yr^{-1}$ (Joughin et al., 2009).  Melt production is concentrated in the fast-flowing lower trunk of the glacier, but is locally larger than 20 mm $yr^{-1}$ even in some regions within the slow-flowing catchment.  Interpretation of radar-reflection properties has also led researchers (Schroeder et al., 2015; Schroeder et al., 2013) to identify an upstream region where water may drain through a persistent,

distributed network of high-aspect-ratio canals, and a downstream region drained by larger canals that concentrate water into a small area. The combination of radar observations and estimated melt rates led to a map of geothermal heat flux, based on the assumption that the basal water system was in equilibrium with steady-state melt rates (Schroeder et al., 2014). Further, the spatial correlation between relatively high driving stress in the lower trunk and the hypothesized channelized drainage system has led to speculation that the character of the basal water system plays a role in the stability of the glacier, and that changes in this water system could lead to accelerated grounding-line retreat (Schroeder et al., 2013).

Active subglacial lakes (lakes that drain or fill over the course of a few years or less) have been identified throughout Antarctica (Smith et al., 2009). Well-documented lake systems have been observed in slow-flowing ice-sheet tributary regions (Wingham et al., 2006) near ice-shelf grounding lines (Fricker et al., 2007), and around outlet glaciers (Stearns et al., 2008). They are commonly associated with fast-flowing glaciers, where meltwater produced by basal sliding is abundant, and where surface and subglacial topography combine to produce hydropotential features that trap water at the bed (Bindschadler and Choi, 2007).

Although active lakes have been identified under many of the large glaciers in Antarctica, none have been found under the Amundsen Coast glaciers (Smith et al., 2009). One explanation for this lack is that the large meltwater production by the fast-sliding glaciers in this region has produced a stable, channelized drainage network that prevents water from accumulating in lakes. An alternate explanation is that the laser-altimetry-based survey that identified lakes in other parts of Antarctica did not make adequate measurements over the cloudy Amundsen Coast to detect the lakes that were there.

To overcome the limitations of the existing altimetry record, and to extend the altimetry record nearly to the present day, here we use CryoSat-2 data to map elevation changes on Thwaites Glacier, West Antarctica (Figure 1). Our results show significant water movement beneath the glacier, and imply temporal variability in water flow that is not captured in published models of the glacier.

## 2. Data and techniques

This study combines surface elevation, ice-thickness, and ice-speed data from a variety of sources. We describe each below.

### 2.1 Surface elevation and elevation-change estimates

The primary source of elevation data for this study are the CryoSat-2 baseline-B radar altimetry data collected between November, 2010 and February 2015, which we refine through a number of steps to derive both POCA (point-of-closest-approach) and swath-mode elevations. With POCA processing, estimates are made of the height and location of the point on the surface that was closest to the satellite when it transmitted a burst of energy, which produces a short-duration, high-energy reflection. Rather

than use one of the level-2 retracked products, we apply a maximum-slope retracker to measure the height of these points (Gray et al., 2015), smoothing each waveform with a Gaussian window with a $\sigma$ of 4 samples, and using a threshold of $3.6\times10^{-16}$ *W sample$^{-1}$* to identify waveform-power slopes significantly above the noise floor, and a coherence threshold of 75%.

Like other POCA products, our retracked POCA CryoSat-2 data tend to preferentially measure the heights of local rises on the ice sheet, missing local depressions, which can lead to gaps in the coverage of measurements of up to a few km in the Thwaites Glacier region. To help fill these gaps, we derived additional elevation measurements using a swath-processing strategy to calculate surface heights from the 'tail' of the return (Christie et al., 2016; Foresta et al., 2016; Gray et al., 2013; Hawley
et al., 2009), which measures energy returned after the POCA. To estimate heights from this part of the waveform, we smoothed the complex phase for each waveform to about 80% of the instrument bandwidth, using a Gaussian kernel with a $\sigma$ of 5 bins, weighted by the coherence values, and calculated a weighted mean of the coherence using the same weights.   For most bursts, this resulted in smoothed coherence curves that were high (greater than 75%) over one or more contiguous segments
after the POCA points. For any segment longer than 20 bins, we unwrapped the smoothed phase starting at the centre of the segment and, geolocated the measurements for three distinct ambiguity shifts of $-2\pi$, 0, and $2\pi$. For each segment, we then fit a spline curve to the derived elevations as a function of the across-track distance to the resulting height values with a resolution of 400 m. For each 400-m node in the spline, we calculated the median height (and across-track offset) and the RDE
(Robust Dispersion Estimate, equal to the half the difference between the 84[th] and 14[th] percentiles) of the residuals to the spline within 200 m of the node. We then compared the median heights to a DEM (based on mosaicked WV DEMs (Shean et al, 2016) and IceBridge altimetry), and selected the best ambiguity that minimized the median absolute difference of the height residuals for each segment. The swath-height values supplied to our fitting routine are the median-residual elevations (and their
locations) around each node, and the errors are the RDEs of the spline residuals.

Swath-processed data over the Thwaites region are affected by metre-scale biases that are correlated over tens of kilometres, and apparently independent from orbit to orbit, possibly because of the time-variable subsurface penetration of radar energy. By combining swath and POCA data, we were able to partially correct for these biases, allowing nearly continuous, dense coverage of our study
area.   We produced surface-elevation estimates with uniform spacing in space and time using a technique that minimizes the misfit between the irregularly sampled data and a smooth surface-height model that varies in time. In this model, the ice-sheet surface is represented as a digital elevation model (DEM) for June 1, 2011, combined with a set of correction surfaces that map elevation changes between the 2011 DEM and the surface for 3-month increments between 2010 and 2014. This technique
minimizes a residual that depends on the roughness of the surface, the spatial variability of the elevation

change rate, and the misfit between the surface and the data (Appendix A).  With this model, we were able to reconstruct the surface height for our region at any time between mid 2010 and late 2014.

## 2.2 WV photogrammetry elevations

5       In addition to CryoSat-2, we derived surface DEMs from WV optical stereo data processed with the Ames Stereo Pipeline (Shean et al., 2016). These DEMs have a horizontal resolution of approximately 12 m; experience with these data suggests that each DEM has a uniform bias of around 4 m (RMS), and that correcting for this bias leaves sub-metre vertical errors that are correlated at sub-kilometre  scales (Shean et al., 2016). While these data have far finer resolution than CryoSat-2, coverage is much more limited.   We constructed a nearly seamless composite DEM for part of our
study area from two overlapping pairs of images, both from November of 2014, by correcting for the mean height bias between the two in their overlap area.   Two earlier image pairs, from November of 2012 and March of 2013, gave estimates of the surface height for part of the same area, approximately two years earlier.

## 2.3 IceBridge elevation differences.

15       NASA's Operation IceBridge program has made extensive laser-altimetry surveys over TWG. We generated a set of elevation-difference estimates spanning the 2013-14 surface drawdown based on ATM surveys in the austral springs of 2010, 2012, and 2014 (Krabil, 2010, updated 2016) and LVIS surveys in the austral springs of 2011 and 2015 (Blair and Hofton, 2010, updated 2016).  We segregated the measurements into an early group, collected between 1 January 2010 and 1 January 2013, and a late
group, collected between 1 September 2013 and 1 January 2015.   For any pair of point measurements in the two groups that were within 100 m of each other, we calculated the elevation difference, using the surface slope estimated from the later of the two surveys to correct for the spatial offset between the two points.   This gave us a collection of elevation-difference measurements that included elevation difference signals over intervals between 2-5 years.   We corrected these elevation differences, first for
the effect of firn-thickness changes using the output of a firn model (Ligtenberg et al., 2011) driven by RACMO2.3 surface-mass-balance estimates (van Wessem et al., 2014), and second for a regional elevation-change rate pattern, calculated from the median of firn-corrected elevation-change rates in 50-meter elevation bins.  The remaining elevation changes show surface-change anomalies relative to the mean regional drawdown pattern.

## 2.4 Bed DEM

      We generated a bed DEM based on the latest available radar-sounding data for the Thwaites region, using a smooth-spline interpolant.   As an alternative, we did consider the Bedmap2 DEM (Fretwell et al., 2013), but did not use it because it does not include the high-resolution 2012 IceBridge

radar-sounding campaign. To derive an estimate of the bed elevation, we used MCoRDS Level-2 (Leuschen et al, 2010) and AGASEA (Blankenship et al, 2012) ice-thickness estimates. Both radar surveys were accompanied by laser-altimetry data sets (Blankenship et al., 2012 , updated 2013; Krabil, 2010, updated 2016), so we converted the thickness estimates to bed-elevation estimates by subtracting them from the mean of all laser altimetry measurements within 100 m of the posted thickness estimate location. We used only those radar-sounding estimates collected when the aircraft was less than 3000 m above the surface, which removes many of the most error-prone measurements.

To interpolate a bed DEM from these data, we used an algorithm similar to that used for the CryoSat-2 surface interpolation (Appendix A), solving for a time-invariant bed elevation model. The bed-fitting algorithm included adjustable parameters that controlled the smoothness and the flatness of the interpolated bed, corresponding to parameters $L_x$ in equation A7 and $w_{x0}$ in equation A8. A set of constrained, independent parameters allowed for a distinct height bias for each day on which data were collected, to account for inconsistencies in radar-system parameters and bed-return picking by different operators. The solution gave bias magnitudes less than 6.5 m, 68% of which are less than 2 m; the standard deviation of the difference between the recovered bed DEM and the data points is 11.3 m.

**2.5 Hydraulic potential mapping**

To help assess the possible water flow at the glacier bed beneath our study area, we mapped the hydraulic potential. This mapping requires surface and bed elevation data.

We estimated the hydraulic potential at the bed of the ice sheet as:

$$\phi' = P_w + \rho_w g z,$$

(1)

Here $P_w$ is the water pressure, $\rho_w$ is the density of water $g$ is the acceleration due to gravity, $z$ is the elevation of the glacier bed above the geoid, and $\phi'$ is the hydraulic potential, in units of pressure. If the basal water pressure is equal to the ice overburden pressure, as is commonly assumed, we obtain the glaciological hydraulic potential (Shreve, 1972), which we divided by the unit weight of water to obtain:

$$\phi = \frac{\phi'}{\rho_w g} = \frac{\rho_i}{\rho_w} z_s + \frac{(\rho_w - \rho_i)}{\rho_w} z_b ,$$

(2)

Here $\rho_i$ is the density of ice, $z_s$ is the surface height of the glacier, and $\phi$ is the hydraulic potential in units of height. We calculated the hydropotential for our field area using the CryoSat-2 surface height estimate for June 1 2011, combined with a bed DEM based on radar sounding data; both are measured relative to the EGM-2008 geoid.

If the water pressure at the bed is equal to the overburden pressure, water at the bed will tend to flow along paths parallel to the gradient of $\phi$, from high to low.    This allows the potential map to define the general direction that water paths are likely to follow.   We used a D8 routing scheme (for 8-directional, meaning that each pixel routes water to the lowest of its eight neighbours) to calculate the

predicted motion of water between nodes in our hydropotential grid (Schwanghart and Scherler, 2014).

At short spatial scales, our potential map contains many locally closed basins. On a bed such as this, long-distance water transport cannot take place unless water can seep through subglacial till, through valleys in the bed too small to be resolved by the radar surveys, or through low-pressure channels that allow water to flow against the local potential gradient.  To represent the large-scale basal

flow pattern, we 'conditioned' our hydropotential map, by artificially filling the closed depressions within each basin to the potential of the lowest point on the boundary, to yield a potential map through which water can steadily flow, because once a basin has been filled, any further water added to it will flow out.  Artificially increasing the potential by 1 m is equivalent to raising the bottom of the ice and the surface together by 1 m, or to raising of the bottom of the ice by about 9 m.  This conditioning had a

similar effect to running a transient water-flow model to steady state before calculating flow paths (Le Brocq et al., 2009) and is among the common strategies used in mapping subaerial flow networks using DEMs that do not resolve the details of every stream valley (Reuter et al., 2009). In some cases, adjacent basins merged during the filling process to form larger, but still closed basins; we continued filling the merged basins until there are no closed contours in the potential map, and used this merged

potential map to derive large-scale flow paths.

**2.6 Ice-surface velocity mapping**

We derived ice-surface velocity maps from a combination of published SAR (Synthetic Aperture Radar) data (Mouginot et al., 2014),  feature-tracked Landsat data between 2012 and 2016, and TerraSAR-X and TanDEM-X (TSX and TDX, referred to generically as TSX) data pairs acquired

between July 2011 and August 2014.

To derive speed estimates from the Landsat data, we used software based on SAR-speckle-tracking algorithms (Joughin, 2002) to estimate offsets between pairs of images and to estimate the contribution of geolocation biases in the images to velocity estimates based on control data on slow-flowing ice. To reduce velocity errors due to geometric distortions in the imagery, we tracked only pairs

of images collected on the same path and row.  This should ensure that errors in feature positions due to inaccuracies in the Landsat geolocation model are nearly the same for both images in a pair, and their errors should cancel in the velocity estimate.

We also derived velocities for twenty-five periods between June, 2011, and August, 2014, from feature tracking in TSX and TDX image pairs (Tedstone et al., 2014) for an area near the grounding line

of Thwaites Glacier.   Since there is no exposed rock or truly stagnant ice in these image pairs, in the

first instance the image coregistration and geocoding is performed using the satellites orbits and a reference DEM; in order to minimize the impact of orbital errors we feature tracked only image pairs with a minimum of 22 days (2 TSX repeat cycles) time-span. To refine the static offsets for each pair, we identified a relatively slowly moving area in the southeast corner of the maps (area C in Figure 6) that was well covered in every epoch, and where we assumed that the ice speed was constant. We subtracted the difference between the speed for this area and its mean speed before June 1 2012 from each speed map. We then calculated velocity anomalies for each corrected speed map relative to the pre-june-1-2012 mean speed map. Corrected speed anomalies are mapped in Figure S5.

## 2.7 Subglacial melt-rate estimates

We also used a map of the estimated subglacial melt rate in our area as derived from the surface velocity map, and an estimate of the basal shear stress derived using inverse methods and an ice-flow model (Joughin et al., 2009). This melt-rate estimate does not include the elevated geothermal heat flux that has been hypothesized based on estimates derived from radar (Schroeder et al., 2014), hence it may underestimate actual melt volumes.

## 3. Results

### 3.1 Ice-surface elevation and elevation change

The derived June 2011 reference DEM for the Thwaites basin does a good job of resolving kilometre-scale features (see the comparison of surface slope and an optical image mosaic in Figure S2). Point-for-point comparison between the DEM elevations and scanning laser altimeter data from 2011 and 2012 shows that the DEM is about 0.6 m higher than the laser data (based on the median difference) with a scatter of about 3 m (based on half the difference between the $16^{th}$ and $84^{th}$ percentiles of the distribution). This comparison suggests that the CryoSat-2-based DEM should provide good estimates of relative height differences across our study area, with local errors on the order of 3 m.

Figure 2A shows the surface elevation change over the 18-month period from June 2011 to January 2013. Overall there was little elevation change over this period. With possible exception of some thinning in the lower left corner, the area shown is far enough upstream that the strong (metres) thinning in response to ice-flow acceleration near the grounding line in the late 1990s and early 2000s (Medley et al., 2014; Mouginot et al., 2014) is not evident. By contrast, Figure 2B shows strong, localized elevation change over the period from January 2013 to June 2014. The most prominent feature in these maps are four oblong-shaped regions where the surface dropped by many metres. The centres of the features are approximately 70, 124, 142, and 170 km upstream of the grounding line, so we refer to them as $Thw_{70}$, $Thw_{124}$, $Thw_{142}$ and $Thw_{170}$. The largest feature, $Thw_{124}$, is roughly oval,

about 16 km wide and 39 km long. Just upstream is $Thw_{142}$, an elongated feature about 20 km east-to-west and about 2 km south-to-north. Farthest upstream is $Thw_{170}$, which extends 11 km south-to-north, and 18 km east-west. The downstream-most feature, $Thw_{70}$, is angular in shape, with the largest drawdown concentrated in a region elongated in the NW-SE direction. A map of the density of elevation measurements remaining after our iterative editing process (Figure S3) shows that while POCA measurements tended to cluster on local highs on the surface while swath measurements are more broadly distributed, points from each of the two sets of measurements contribute to elevation estimates within the outlines. This shows that both types of data contribute to the measured elevation changes, and that the elevation differences are not solely due to bias changes in the swath-processed data.

An independent set of measurements from the WorldView-2 (WV-2) satellites and Operation Ice Bridge altimetry shows the elevation-change pattern for a portion of the western sides of $Thw_{124}$ and $Thw_{70}$. The November-2014 WV-2 DEM shows the surface height after the surface change was largely finished. We subtracted the heights measured in the two earlier DEMs, from November 2012 and March 2013, and corrected for residual biases and regional elevation change by subtracting the mean elevation difference outside the boundaries of $Thw_{124}$ and $Thw_{70}$. We combined this with IceBridge elevation differences, corrected for the regional draw-down pattern and firn–thickness change. The resulting elevation-change maps (Figure 2C) show small (~0.7 m RMS) apparently random elevation variations outside the feature boundaries, up to 20-m subsidence at $Thw_{124}$, around 6 m drop at $Thw_{70}$ and $Thw_{142}$, and about 2 m drop at $Thw_{170}$. The spatial patterns and magnitudes of these changes are similar to those measured by CryoSat-2.

To derive a volume change for the features, we followed a procedure similar to that used in previous studies of active subglacial lakes (Flament et al., 2014; Fricker et al., 2007; Smith et al., 2009). We drew a bounding polygon for each feature that encompasses all substantial (> 0.5 m) elevation change. For each three-month elevation-difference surface, we subtracted the elevation change within the polygon from the elevation change in a region between 2 km and 6 km outside the polygon, which corrects for large-scale elevation-change errors, as well as regional drawdown associated with ice-dynamic thinning. The top panel of Figure 3 shows the mean elevation change with time for each feature. Integrating these corrected changes in space gives the volume change for each feature. Both the elevation and the volume change (Bottom panel of Figure 3) show nearly concurrent drainages. It appears that $Thw_{124}$, began to deflate first, (January 2013) and continued losing water until mid 2014, with a total loss of 3.7 $km^3$. In March 2013, $Thw_{170}$ began to drain, and continued until the beginning of 2014, with a total loss of 0.49 $km^3$. Third in this progression was $Thw_{142}$ draining 0.54 $km^3$ between June of 2013 and January of 2014. Finally, $Thw_{70}$ lost 0.87 $km^3$, somewhat more slowly, but primarily between June 2013 and June 2014. The timing of each of these events is somewhat uncertain, because the season-to-season coverage by CryoSat-2 of each feature is inconsistent, and the smoothing

constraints applied during the fitting process are expected to yield elevation-change estimates that are temporally smoother than the actual pattern of elevation change; this latter effect is particularly strong at $Thw_{124}$ because the smoothing constraints tend to blur the timing of very large changes. As a result, we cannot rule out the possibility that the elevation changes happened at the same time for all four features, or in a different sequence than just described, although the data, taken literally, appear to indicate an earlier change at $Thw_{124}$.

## 3.2 Hydropotential maps

Figure 4A shows the hydropotential map derived for our study area. The surface elevation variations are largely responsible for determining the potential basin shapes, and define distinct basins for each feature. The exception to this pattern is $Thw_{70}$, which spans a range of hydraulic potentials, with its downstream end about 200 m lower than its upstream end. Regionally, there is a strong potential gradient driving water parallel to the ice-flow direction, which means that the upstream features have higher potentials than the downstream features. We take the mean potential of the digitized boundary of each feature as representative of the height of the boundary controlling flow into or out of the feature. From upstream to downstream they are 1000, 930, 855, and 715 m, respectively. The differences between the hydropotential map derived from our spline-fit DEM and the Bedmap2 DEM (Fretwell et al., 2013) are small, suggesting that our analysis does not depend strongly on the choice of the bed DEM used (see Figure S4 for a comparison).

Figure 4B shows the potential difference required to achieve the connected potential map. The merging process (see methods) produced one main basin for each of our drainage features, except for $Thw_{70}$, which is divided among four. For most basins, the potential difference is 1-2 m; but for a few, including those associated with $Thw_{124}$, $Thw_{142}$, and $Thw_{170}$, the potential change required was on the order of 10-20 m. There is also a large area just downstream of, and parallel to, $Thw_{124}$ that in places required filling by more than 30 m. Figure 4D shows the flow paths calculated from the filled potential map. This map includes a path that skirts the eastern edges of $Thw_{170}$ and $Thw_{142}$, passes through $Thw_{124}$ from east to west, then sweeps to the northwest, missing $Thw_{70}$ entirely, and meeting the grounding line approximately in the centre of the fastest-flowing part of Thwaites Glacier. The flow paths and basins generated with the Bedmap2 DEM are qualitatively the same over most of the domain, although it shows $Thw_{170}$ draining to the west into a channel that bypasses $Thw_{142}$ and $Thw_{124}$, and connects to the drainage from $Thw_{124}$ just downstream of that lake (see Figure S5).

The melt-rate map (Figure 4C), combined with the (merged) basin map allow an estimate of the melt-water supply rate to each of our features. First, for each feature, we calculated the rate of melt production within the feature's local catchment. Next, with the large-scale drainage map, we calculated the rate of melt production over the entire catchment for the feature. These volume rates are reported in Table 1.

### 3.3 Ice-velocity mapping

A profile of speeds for a flowline that runs through our features and out onto the central flowline of the glacier is shown in Figure 5, as well as a set of interpolated speeds for a point just downstream of $Thw_{70}$, and at the grounding line. The most prominent features in these plots are the long-term acceleration of Thwaites Glacier, which has a large effect near the grounding line but much smaller 70 km upstream, and scatter in speeds between different sensors, which produces substantial, but not meaningful, apparent speed differences in the velocity-profile plots and in the time-series plots. Any speed change associated with the lake drainage is small compared to the decadal-scale speed variations of the glacier.

A more self-consistent and detailed estimate of speed change around the time of the drainages comes from a set of TSX velocity maps near the grounding line in the fastest part of the glacier (Figure 6, supplemental material Figure S5). These maps show that a small area, about 15x20 km in extent, on the east side of the glacier, accelerated by about 100 m $yr^{-1}$ over the course of the 2013 calendar year, then slowed by about half as much over the course of 2014. By contrast, the ice 20 km to the west of the main trunk slowed at about 50 m $yr^{-2}$ until the start of 2013, then maintained an approximately constant speed through the end of 2014. The centre of the acceleration feature is within a few kilometres of the drainage path inferred from the hydopotential maps.

### 4. Discussion

One possible explanation for the observed changes is that they reflect changes in surface properties and their interaction with CryoSat altimetry measurements. Near-surface density can vary in time (Ligtenberg et al., 2011), and these variations are can cause both real surface-elevation changes and apparent surface-elevation changes due to changes in the penetration of radar altimeters' energy into the firn (Ligtenberg et al., 2012). At the same time, firn density likely varies on short spatial scales on Thwaites glacier, driven in part by surface slope variations (Grima et al., 2014). These two effects together might lead to apparent surface-elevation changes in CryoSat data, on the spatial scale of the changes observed here. We believe that these effects played at most a minor role in the changes we observed. The close agreement between the surface-elevation changes measured by CryoSat, laser altimetry, and photogrammetry in the areas where they overlap suggests strongly that the CryoSat changes reflect real changes in the surface height, and not temporal changes in subsurface penetration of radar energy. Given that the surface elevation likely changed by several meters, it seems unlikely that changes in firn density alone could have produced these changes. The total range of estimated firn-air content change for this area between 1979 and 2012 based on firn modelling driven by reanalysis data is less than 1 m (Ligtenberg et al., 2014), much smaller than the 4-20 m changes observed here.

Following previous studies of similar features (Fricker et al., 2007; Gray et al., 2005; McMillan et al., 2013; Smith et al., 2009; Wingham et al., 2006) the simplest and most likely explanation for the observed changes in surface height is the sudden drainage of four subglacial lakes, and we will hereafter refer to the features as lakes. Although some previous studies (Smith et al., 2009) have recommended

caution in using coincident filling or draining of adjacent lakes as evidence of hydraulic connection, the nearly simultaneous drainage of four lakes strongly suggests some kind of linkage in the basal hydrological system. Detailed modelling of the surface changes associated with changes in basal topography (Gudmundsson, 2003; Sergienko et al., 2007) show that in fast-flowing environments, ice flow changes in response to perturbations in the surface shape can reduce the amplitude of surface

elevation change in response to changes at the bed. Specifically, a lake that drains at the bed will produce a surface depression, but ice flowing into the depression will quickly reduce its depth. The net volume of the ice sheet must be conserved, so that the volume of the depression at the surface must equal volume drained at the bed, but as ice flow refills the lake depression, and the correction we make for regional uplift or drawdown could lead us to an overall underestimate of the lake volume change.

This suggests that the volume of water displaced at the glacier bed during the lake drainages was larger than the volume of the changes at the surface, and that our measurements represent a minimum estimate of the water movement. Lacking any technique for estimating the relationship between the two volumes, we proceed as if they were equal, but acknowledge that there is uncertainty in this approximation. By contrast, changes in basal drag (i.e. the appearance or disappearance of sticky spots)

can produce changes in surface topography, but these changes should appear as dipole-like patterns oriented in the along-flow direction, with no net volume change (Gudmundsson, 2003). We do not see evidence of this kind of pattern in our altimetry measurements.

**4.1 Basal Hydrological System: Linked Lake Catchments**

The hydropotential mapping shows that subglacial water flow beneath Thwaites Glacier is

organized by surface topography into circuitous paths that are often perpendicular to the large-scale flow gradient. The cross-slope water paths are defined primarily by elongated ridges in the glacier surface. Although the bed topography plays a lesser role in defining the flow directions, surface undulations often are muted expressions of features at the bed (Gudmundsson, 2003). Previous studies (Bindschadler and Choi, 2007; Siegert et al., 2014) have identified locations such as these as likely to

trap water, and have shown that even on smooth beds, surface topography generated by local variations in basal traction can produce hydropotential basins that trap water (Sergienko and Hulbe, 2011).

Our analysis of the hydropotential maps suggests that together the interaction of bed and surface topography produces a basal hydrologic system that consists of many individual catchments, linked by a series of drainage paths that are at least intermittently active. The bumps at the surface that give rise to

the catchments represent large excursions in the driving stress, which is associated with a locally

elevated meltwater supply for each catchment. These factors together create an environment favourable to the accumulation of water to form subglacial lakes. Many features do not represent deep sinks in the hydropotential map, so they may simply collect water in a region with little storage, which then cascades downstream to the next catchment. Some of these features, however, represent much deeper sinks in the potential field, which can allow lakes with substantial volume to fill and drain.

Our hydropotential map is not a perfect tool for predicting water flow at the bed since it makes the assumption that the water pressure equals the overburden (i.e., zero effective pressure). Since most sliding laws produce zero resistance with zero effective pressure, at least in some regions the water pressure must be lower than assumed to maintain basal traction, particularly beneath regions where surface slopes are steep (i.e., driving stresses are high). Thus, there is uncertainty in our estimates of the details of hydropotential that reflect features and processes that we are unable to resolve with the present data. As a result, while we cannot precisely determine the nature of each flow path from the data, it does appear that some lakes may be connected continuously, while others may have more intermittent connection. In the latter case, only when the lakes have filled such that their potential exceeds the minimum local hydropotential barrier do they drain. Note the initial drainage might be slow and inefficient, but once started, a low pressure-gradient channel may develop that leads to more rapid drainage. Once the drainage is complete, without water flow to sustain melting, such a tunnel would close and reseal the lake, allowing it to recharge. Despite evident limitations in our hydropotential maps' ability to predict water movement, it appears reasonable to assume that lakes form where sinks in the map are the greatest. In fact, the lake drainages we observe occur precisely where we find some of the deepest closed basins in the hydropotental field (Figure 4B). In areas where catchments are connected more continuously without abrupt drainage, water may either move continuously between catchments either through a small network of tunnels or through a less efficient distributed network. Over much of the area around the lakes, characteristics of radar returns from the bed have led researchers to infer the presence of a basal drainage system comprised of elongated channels running parallel to ice flow (Schroeder et al., 2015; Schroeder et al., 2013). Such a system of elongated canals could prevent the accumulation of large volumes of water if it were broadly connected, so our results suggest that if a large-scale canal system is present around the lakes, there may be gaps in its spatial connections, or it may not have sufficient conductivity to prevent large lakes from accumulating.

The four lakes appear to have drained nearly concurrently, with $Thw_{124}$ appearing to precede the others. The upper three lakes are linked by a through-going potential drainage pathway (Figure 4D), while $Thw_{70}$ ties into this drainage pathway farther downstream. As computed, the drainage pathways only exist when the water level rises such that it overcomes the hydropotential barrier. With this model, an upstream lake could overflow into a downstream lake, which would subsequently cause it to overflow, which would then trigger the next event, a mechanism that has been proposed to explain temporal patterns in surface change in several glacier systems (e.g. Whillans Ice Stream (Fricker et al.,

2007), Recovery Glacier (Fricker et al., 2014), Macayeal Ice Stream (Carter et al., 2011), and in Wilkes Land, East Antarctica (Flament et al., 2014)). While this scenario could produce nearly simultaneous drainage, it is inconsistent with the observations, which suggest that $Thw_{124}$ likely drained first. In principle, the drainage of $Thw_{124}$ should not trigger the drainage of the upstream lakes by the overflow mechanism, in that they would have to exceed their own potential barriers first. Lowering the potential of $Thw_{124}$, however, would have forced more water flow toward the lake, at least within the upper confines of its own catchment. The development of a lower-pressure conduit near the boundary between basins could have altered pressure gradients sufficiently to allow water from the adjacent catchments to spill over, which, through a similar lowering of potential, could have induced drainages of the catchments farther upstream. Perhaps owing to noise in the data or irregular topography, the $Thw_{70}$ basin is made up of several catchments, some of which are nearly adjacent to the large drainage pathway for the other draining lakes. If this path were closed prior to drainage, but opened to accommodate the $Thw_{124}$ drainage, then the increased pressure gradient between the channel and $Thw_{70}$ may have been enough to activate its drainage pathway.

As just described, lowering the hydropotential gradient at the lower end of a drainage pathway may be sufficient to open it for efficient drainage. This is not a completely satisfying explanation, as some of the pathways are quite long. Nevertheless, it is important to keep in mind the actual water pressure distribution is unknown and evolving with time. Thus, some of the lake may be connected by substantially shorter paths than shown, with weaker than indicated potential barriers dividing them. Further explanation into the nature of the triggered drainage likely will require a far more detailed hydrological data, likely constrained by a better resolved bed model.

This picture of lakes and subglacial hydrology complicates the modelling of subglacial water flow. Some techniques for estimating subglacial water flow rates (Schroeder et al., 2014) infer the hydraulic conductivity of the glacier bed under the assumption that the conductivity is sufficient to evacuate the meltwater produced steadily by the glacier. Our results show that the instantaneous conductivity at any time may be substantially too small to evacuate the steady meltwater production upstream, but that while lakes are draining, the conductivity increases dramatically. Over the course of multiple lake-drainage cycles, the time-averaged conductivity should be adequate to remove the steady-state melt, but the balance cannot be assumed at any given moment.

From the melt rate estimates and our inferred drainage pathways, we can make some estimates about the recharge times of the lakes. The last two columns of Table 1 show the time required for each lake to refill after its observed drainage, based on local and on catchment-scale melt production, ranging from 39 to 83 years for the upper three lakes. If the lakes collect water from upstream catchments, however, this range becomes 4.7 to 22 years. As noted above, the melt estimates assume a fairly low estimate of the geothermal heat flux (Joughin et al., 2009), and the actual value could be significantly higher (Schroeder et al., 2014). As a result, these times could be a few years faster than indicated. The

fact that the hydropotential barriers seem low for many catchments favours aggregation of water in a few lake basins with the correspondingly faster recharge times. This is consistent with the relatively abundant observations of active lakes around Antarctica (Smith et al., 2009).

The routing map, combined with the discharge estimates allows us to estimate the rate of water delivery to the lower trunk of the glacier. Given the uncertainty in the timing and magnitude the discharge, we can only offer a lower bound for this, because the altimetry analysis tends to produce a temporally smoothed estimate of surface change, corresponding to a peak volume-change rate that is likely too small. If the water discharged from the lakes followed parallel paths to the grounding line, then the rate of water delivery is equal to the sum of the discharges of the four lakes, with a peak rate of about 7.5 $km^3 yr^{-1}$. If the water from $Thw_{142}$ and $Thw_{170}$ reached the grounding line through $Thw_{124}$, then the rate is equal to the sum of the rates of volume lost by $Thw_{124}$ and $Thw_{70}$, with a peak of about 6 $km^3 yr^{-1}$. In either case, the peak happened over the second half of 2013.

## 4.2 Influence of Drainage on Glacier Speed

The sudden injection of a large volume of water under the trunk of an active glacier has in some cases led to a short-term acceleration in flow and discharge (Stearns et al., 2008). For Thwaites Glacier, however, the extra water seems to have had little or no influence on the speed of the lower trunk of Thwaites Glacier. The largest acceleration detected at the grounding line, during the peak drainage period, amounted to at most 125 m $yr^{-1}$, or less than 10% of the pre-acceleration speed. This is only moderately larger than the longer-term ice-stream speed trend of around 4% $yr^{-1}$ between 2003 and 2010. Moreover, speedups of this magnitude can also be explained by ungrounding in response to ocean melting (Joughin et al., 2014).

The lack of a strong acceleration in response to the lake drainage should not be surprising. The discharge of $Thw_{124}$ only lasted a few months, so even if it had produced a significant ice-speed change, its effect on the net discharge of the glacier averaged over several years would have been minimal. Further, model-based estimates of the basal shear stress of the lower trunk of Thwaites Glacier (Joughin et al., 2009) shows basal drag concentrated in narrow (~5 km wide) bands oriented perpendicular to flow. It seems likely that the glacier speed is largely determined by the drag in the high-stress regions and by lateral shear stress supported outside the lateral margins. If the water discharged from $Thw_{124}$ moved through narrow channels, it would have occupied only a small area of the bed, and the total change in force on the glacier, proportional to the product of the area occupied by the channels and the mean shear stress over that area, would likely have been small. Moreover, channels with lower pressure than the surrounding hydrological system might actually withdraw water from higher-pressure distributed systems and act to decrease speeds. Our results are largely in agreement with the hypothesis that water in the lower part of Thwaites Glacier can travel through channels (Schroeder et al., 2013), but the pre-drainage retention of water suggests that the channels are at most intermittently active. If the

upstream lakes were briefly connected by a low-pressure channel, the lack of substantial glacier slowdown after the end of the subglacial flood suggests that the induced transition from a high-pressure distributed water system to a low-pressure channel was not permanent, or at least that it did not produce a substantial change in basal traction on the glacier.

5  ## 5. Conclusions

Our altimetry measurements reveal a substantial ($3.7$ km$^3$) short-term transfer of water across the bed of Thwaites Glacier.  Multiple subglacial lakes appear to have drained, with a temporal pattern that suggests linkage over more than 100 km, with a pattern of drainage suggesting that the lakes were connected to a subglacial water system that could change its discharge rate drastically over a few 10  months.  Although this water likely reached the fastest flowing part of the ice stream at a flow rate of between 5 and 7.5 km$^3$ yr$^{-1}$, the added water appears to have had no substantial effect on the ice speed, which is different than what has been reported for some other glaciers (Stearns et al., 2008), but not surprising based on principles of basal-hydrological and basal-sliding theory.

Historically, most full ice sheet models have been developed at resolutions of 10 to 40 km, 15  which is insufficient to resolve topography at the scale that gives rise to the linked catchments shown in Figure 4. Most models have assumed relatively smooth gradients in the hydropotential field that drives an efficient or inefficient drainage network, which is generally driven by bed properties at that scale of mm to m. As we are able to measure the ice sheet surface and bed at ever improving resolution, it is becoming apparent that the routing of basal water is highly dependent on processes acting at the km 20  scale and a linked catchment system represents a different paradigm than has or could be considered in most ice sheet models thus far.

While our data suggest water is routed in ways not presently accounted for in most ice sheet models, it also indicates that changes of this type in the basal hydrological system may not matter much. The basal water system is able to sequester large volumes of water over years which it then releases 25  rapidly with little or no apparent change in glacier speed. This insensitivity suggests that the details of the basal hydrological system may not be the most important feature of the ice sheet for models to capture, especially now that data assimilation techniques allow us to infer the dynamic properties of the bed (e.g., the coefficients in a sliding law) directly (Joughin et al., 2010; Morlighem et al., 2010). At least at the decadal scale, fixed bed parameters can reasonably reproduce observed behaviour (Joughin 30  et al., 2010; Joughin et al., 2014),  despite large increases in water-layer thickness that accompany a speedup and lake drainages. The lack of sensitivity is probably related to the patchy structure of basal drag beneath TWG, and the limited time over which lake drainages supply water.  As previous studies have noted (Joughin et al., 2009; Schroeder et al., 2013; Sergienko and Hindmarsh, 2013) much of the drag restraining the ice flow is concentrated in small patches or bands, and if changes in water pressure 35  reduce the drag in the low-drag areas between these patches, the speed of the glacier is unlikely to

change significantly. Further, a short-duration drainage, even of a large volume of water, cannot cause a large change in the long-term average discharge of a fast-flowing glacier like THW. With only a few examples of changes in water availability to Antarctic glaciers documented, data are too sparse at present to say definitively whether an evolving hydrological system is an essential part of a predictive

ice sheet model. Nevertheless, the data that do exist suggest that such sensitivity to hydrological evolution may be small. Existing satellites such as CryoSat, ICESat, and several SAR missions have already provided a wealth of data to explore such issues. The launch of ICESat-2 (Ice, Cloud, and Land Elevation Satellite-2) in late 2017 or early 2018, and the launch of the NASA ISRO SAR (NiSAR) in 2020 will improve this situation considerably.

**Appendix A. Methods for estimating elevation and elevation change.**

Based on the Cryosat data, we estimated elevation changes and a DEM on overlapping 65-km rectilinear grids. Each grid has one set of nodes defining reference DEM heights (for June 1 2011), spaced 500 m in each direction, and one set of nodes defining elevation-change surfaces for 3 month increments between June 1, 2010 and March 1, 2015, spaced at 1 km in each direction. Collectively,

the heights of these nodes constitute an elevation model, giving the height of any point within the grids, for any time between the first and last elevation-change surfaces. The centres of individual grids are spaced every 25 km, so each grid overlaps its neighbours by 20 km. When the solution is complete, the grids are merged into a master grid using a raised-cosine-taper weighting function that ensures that the master grid elevations and elevation changes are smooth across the grid boundaries.

We solved for the surface heights and elevation changes by minimizing a penalty function, $R^2$, that depends on the mismatch between the elevation model and the data, and on the spatial gradients in the maps. Selecting a model (a set of surface grids and a set of bias parameters) that minimizes $R^2$ gives the smoothest model consistent with the data, subject to the choice of trade-off parameters that determine the smoothness of the final model. This penalty function is:

$$R^2 = \sum_{i=1}^{N_{data}} \left( w_i \frac{z_m(x_i,y_i,t_i)+b_i-z_i}{\sigma_i} \right)^2 + w_{x0}F_x(z_0) + w_{xt}F_x\left(\frac{\partial \delta z}{\partial t}\right) + w_{tt}F_{tt}(\delta z) + F_b(b) + F_{\delta z0}(\delta z) \quad ,$$
(3)

The first term is minimized by reducing the data misfit, equal to the difference between the sum of the surface model, $z_m(x_i, y_i, t_i)$, the bias model, $b_i$, and the measured elevations, $z_i$. The other terms are model constraints that impose a penalty on models that have large slopes or roughness, or that have

excessively large biases. The second and third terms are minimized by reducing spatial variations in the DEM height and in the elevation-change rate, fourth term is minimized by reducing the temporal variation in the elevation-change rate at each node, and the last term is minimized when the bias-model parameters are small. Here $F_x$ is an operator that increases with the first and second spatial derivatives

of its argument. $\sigma_i$ are the estimated data errors, and $w_i$ are a set of data weights. The parameters $w_{xo}$, $w_{xt}$, and $w_{tt}$ determine the importance of variations in the spatial and temporal derivatives of the model heights, relative to the other residual errors. $F_b(b)$ is a function of the bias-model parameters ($b$) that increases with their magnitude and/or roughness and curvature. The last term specifies that the elevation-change maps should equal zero for the time step corresponding to the reference DEM.

The surface model is expressed as a set of nodal values for a DEM, and for a set of quarter-annual correction surfaces. The elevation of any point within the model domain, with spatial coordinates $(x,y)$ at time $t$, can be found by spatial interpolation between the DEM nodes to give reference DEM height, and by spatio-temporal interpolation into the elevation-difference nodes to give the height difference between the surface June 1, 2011, and the surface at $(x,y)$ and time t:

$$z_m(x, y, t) = I_{xy}(x, y; z_0) + I_{xyt}(x, y, t; \delta z) \tag{4}$$

Here $I_{xy}$ and $I_{xyt}$ are operators that interpolate the nodal values to the specified locations. We use a bilinear interpolation in space, and a cubic-spline interpolation in time; since these are linear operations, we make this calculation using a matrix multiplication:

$$\mathbf{z_m} = \mathbf{I_{xy;t}} \, \mathbf{m_z} \tag{5}$$

Here $\mathbf{z_m}$ is a vector of heights interpolated from the model and $\mathbf{I_{xy;t}}$ is a matrix that, multiplied by elevation model $\mathbf{m_z}$, gives the height estimates at $x$, $y$, and $t$.

The bias model has one variable for each orbit, $n_k$ that gives a bias for swath elevations, one parameter for all points, $k_p$ that gives the height sensitivity to the log of the returned power, and a 2-km geographic grid of values, $k_{sp}$ that gives a bias between swath and POCA elevations:

$$b_i = n_k \delta \phi_i + k_p(\log(p_i) - \log(p_0)) + \begin{cases} I_{xy}(x_i, y_i; k_{sp}) & for \; swath \; points \\ 0 & for \; POCA \; points \end{cases}. \tag{6}$$

As before, $I_{xy}$ is the operator giving the linear interpolation of the grid of $k_{sp}$ to the measurement points $(x_i, y_i)$. We can write this as a matrix multiplication:

$$\mathbf{b} = \mathbf{Bm_b}. \tag{7}$$

Here $\mathbf{B}$ is a matrix calculated based on the power, phase, and location of the data points, $\mathbf{b}$ is a vector of bias values calculated from the bias model, and $\mathbf{m_b}$ is a vector containing $n_k$, $k_p$, and $k_{sp}$.

Using (5) and (7), we can write the first term of (3) as

$$(\mathbf{G_d m} - \mathbf{z})^T C^{-1}(\mathbf{G_d m} - \mathbf{z}) \tag{8}$$

Here $\mathbf{G_d}$ is the horizontal catenation of $\mathbf{I_{xy;t}}$ and $\mathbf{B}$, and $\mathbf{m}$ is the vertical catenation of $\mathbf{m_z}$ and $\mathbf{m_b}$. **C is a diagonal matrix whose elements give an estimate of the squared magnitude of the uncorrelated component in the data errors, scaled by a weighting factor that attempts to reduce the effects of outlying values on the inversion.** The remaining terms of (3) help select models that have smoother

DEMs, simpler patterns of elevation change, and less complicated bias models. The operator $F_x$ is a discrete approximation of the function

$$F_x(z) = \int \left(\frac{\partial^2 z}{\partial x^2}\right)^2 + 2\left(\frac{\partial^2 z}{\partial x \partial y}\right)^2 + \left(\frac{\partial^2 z}{\partial y^2}\right)^2 dA + \frac{1}{L_x^2}\int \left(\frac{\partial z}{\partial x}\right)^2 + \left(\frac{\partial z}{\partial y}\right)^2 dA \tag{9}$$

When applied to the elevation-change maps, this operator is summed over all pairs of subsequent surfaces. The value of $L_x$ determines the relative importance of the model gradients and the model curvature to the total residual; it gives the approximate distance over which the surface slope in an unconstrained part of the model approaches zero. We set it to 1 km, the approximate size of gaps between POCA points from distinct tracks in our study area. Discretizing this operator lets us write the second and third terms of (3) as

$$w_{x0}(\mathbf{F_0 m})^T \mathbf{F_0 m} + w_{xt}(\mathbf{F_\delta m})^T \mathbf{F_\delta m} \quad . \tag{10}$$

Here $\mathbf{F_0}$ is a discretized version of the gradient of (10) as applied to the DEM, and $\mathbf{F_\delta}$ is a discretized version of the gradient of (10) as applied to the elevation-change maps (i.e. the difference between subsequent $\delta z(t)$ maps).

The fourth term of (3) is minimized by reducing the temporal variation in the rate of elevation change at each node in the $\delta z(t)$ maps. $F_{tt}(\delta z)$ approximates:

$$F_{tt}(\delta z) = \iint \left(\frac{\partial^2 \delta z}{\partial t^2}\right)^2 dt\, dA. \tag{11}$$

This operator is discretized on the nodes of $\delta z(t)$, allowing us to write it as:

$$(\mathbf{F_{tt} m})^T \mathbf{F_{tt} m}. \tag{12}$$

Here $\mathbf{F_{tt}}$ operates only on the elements of $\mathbf{m}$ corresponding to $\delta z$.

The fifth term of (3) minimizes the magnitude of the bias model:

$$F_b = w_{sp} F_x\left(k_{sp}\right) + \sum_{i=1}^{N_{orbits}} \left(\frac{k_{pi}}{k_{p0}}\right)^2 \quad . \tag{13}$$

The first term of (13) contributes a larger penalty for larger swath-POCA biases, the second term contributes a larger penalty for larger phase-dependent biases. In matrix notation, the fifth term of (3) is:

$$(\mathbf{F_b m_b})^T \mathbf{F_b m_b}. \tag{14}$$

The last term of (3) is used to force the elevation increment for June 1 2011 to be equal to zero. This effectively specifies the date for the DEM:

$$F_{\delta z0} = W \sum_{i \in june\ 2011} \delta z_i^2 \tag{15}$$

Here W is an arbitrary weight, which we set to a large enough value that the elevation difference values for June 1 2011 are less than 1 mm. The matrix form of (15) is $(\mathbf{F}_{\delta z0}\mathbf{m})^T \mathbf{F}_{\delta z0}\mathbf{m}.$

To solve for the elevation model and bias parameters, we find a model that minimizes R by solving for elevation- and bias-model variables that make the derivative of (3) with respect to the model parameters equal to zero. This leads to a set of linear equations:

$$
\begin{bmatrix} \mathbf{G_d} \\ \mathbf{F_0} \\ \mathbf{F_\delta} \\ \mathbf{F_{tt}} \\ \mathbf{F_b} \\ \mathbf{F_{\delta z0}} \end{bmatrix} \begin{bmatrix} \mathbf{m_0} \\ \mathbf{m_\delta} \\ \mathbf{m_b} \end{bmatrix} + \epsilon = \begin{bmatrix} \mathbf{z} \\ 0 \\ 0 \end{bmatrix} \tag{16}
$$

Here $\mathbf{z}$ is a vector of surface-height estimates. We solve this by minimizing the quantity $\epsilon^T \mathbf{C}^{-1}\epsilon$, where $\mathbf{C}$ is a matrix whose diagonal values give the weights for each component of (3). In principal, this could be solved by standard linear-least-squares techniques (Menke, 1989) but because of the large number of equations and unknowns, we use the Matlab routine *lscov*, which uses an algorithm designed to efficiently solve large, sparse systems of least-squares equations.

We select weights for our data residuals using the iteratively reweighted least-squares technique (Osbourne, 1985) with a Tukey weighting scheme with a threshold parameter of 3: We calculate the solution initially setting $w_i=1$, then recalculate the weights based on the residuals between the model and the data:

$$
w_i = \begin{bmatrix} 0 & \left|\frac{r_i}{\sigma_i}\right| > 3\max(1,\hat{\sigma}) \\ \left(1 - \left(\frac{r_i}{\sigma_i max(1,\hat{\sigma})}\right)^2\right)^2 & \left|\frac{r_i}{\sigma_i}\right| \le 3\max(1,\hat{\sigma}) \end{bmatrix} \tag{17}
$$

$\hat{\sigma}$ is a robust estimate of the spread of the scaled residuals with nonzero weight from the previous iteration:

$$
\hat{\sigma} = \frac{1}{2}\left(P_{84}\left(\frac{r'}{\sigma_z}\right) - P_{14}\left(\frac{r'}{\sigma_z}\right)\right) \tag{18}
$$

Here $P_{84}()$ and $P_{14}()$ are the 84[th] and 14[th] percentiles of the distribution of the quantity in parentheses. By construction, $\hat{\sigma} = 1$ for a normalized Gaussian distribution, but outlying residuals affect $\hat{\sigma}$ less than they would the standard deviation. As we repeat this process over multiple iterations, outlying data are assigned smaller and smaller weights, and the solution converges until either the smallest difference between δz values for two subsequent iterations is less than 0.01 m, or until 20 iterations are complete.

One complication in the iterative-fit procedure is that data with elevations tens of metres from the true surface can produce 'spikes' in the DEM that slow the convergence of the entire system. To help eliminate these, when, for a given iteration, the second derivative magnitude for a point in the DEM is greater than $10^{-4}$ m$^{-1}$, all data within 1 km of that point are removed from the solution at the

start of the next iteration. At the end of the next iteration, the solution around the point is usually much smoother, the erroneous data are treated as outliers (with $r>3\hat{\sigma}$) in subsequent iterations, and the remaining, non-outlier data around the point are used in the solution.

The selection of the weighting parameters $w_{x0}$, $w_{xt}$, and $w_{tt}$ is carried out through a combination of arbitrary choices and hand tuning. The initial values for each parameter are set based on reasonable values for an ice-sheet, using the formulas:

$$w_{xo} = \left[A\, E\left(\frac{\partial^2 z_0}{\partial x^2}\right)^2\right]^{-1}$$

$$w_{xt} = \left[A\, E\left(\frac{\partial^3 \delta z}{\partial x^2 \partial t}\right)^2\right]^{-1} \tag{19}$$

$$w_{tt} = \left[A\, E\left(\frac{\partial^2 \delta z}{\partial t^2}\right)^2\right]^{-1}$$

Here A is the domain area and E() is the expected value for a quantity. The normalization ensures that if each quantity in the model is equal to its expected value, the corresponding term in (3) is equal to unity. We began by exploring a range of parameters around the values listed in Table 2. The centre of the range for $w_{x0}$ was chosen based on surface topography with an amplitude of 50 m at a wavelength of 6 km (typical values in an ice-stream environment), the range for the elevation-rate variability parameter is centred on a value chosen based on an 0.1 m yr$^{-1}$ variation in the elevation-change rate on a 5-km wavelength. The centre of the range for $w_{tt}$ was chosen based on snow-accumulation-rate variability in the Thwaites catchment, on the order of 1 m yr$^{-2}$. For each parameter we tested expected values within 1-2 orders of magnitude of the centre of the range and evaluated whether each value allowed the inversion procedure to reject outlying data points, while still capturing the pattern of elevation change around Thw$_{124}$ (Figure S1). The solution is relatively insensitive to variations in $w_{tt}$ and $w_{x0}$, with variations around the chosen value by a factor of 30 producing only minor changes in the recovered pattern of elevation change and the DEM shape. Increasing $w_{tt}$ by more than a factor of 100 (i.e. seeking a much smoother solution in time) resulted in more severe data editing, and began to degrade the spatial sampling of the solution. By contrast, decreasing $w_{tt}$ by a factor of 10 resulted in a much rougher $\delta z$ field, while increasing it by a factor of 10 resulted in a blurred map of $\delta z$. With our chosen values, the iterative weighting scheme had nonzero weights for about 90% of input data points, and returned a $\hat{\sigma}$ value of 1.06 m.

A shaded-relief map of the June 1, 2011 surface DEM derived using the selected weights is shown in Figure S2. To demonstrate the accuracy of this result, we also show a subset of an optical-image mosaic of Antarctica for the same area. We adjusted the shading azimuth and elevation to achieve a best match between the two, but comparing these maps shows that the DEM captures the few-kilometre-scale surface topography that is visible in the image mosaic. A map of the density of POCA and swath elevation measurements after the iterative data editing is shown in Figure S3.

**Author Contributions.**

Smith and Gourmelen performed initial altimetry research. Smith developed transient elevation-change and DEM algorithms, and coordinated overall data analysis. Joughin produced Landsat-velocity and melt-rate estimates. Gourmelen produced SAR velocity estimates. Huth developed Cryosat analysis
software. All authors contributed to manuscript writing and editing.

**Acknowledgments**. Work on this paper was funded by NASA grant NNX13AP96G (BS and AH), NSF grant ANT-0424589 (IJ) and European Space Agency's Support to Science Element programme through CryoTop project 4000107394/12/I-NB and CryoTop Evolution project 4000116874/16/I-NB
(NG). DLR project gourmele_XTI_GLAC0296 provided the TSX data. We Acknowledge the support of the Polar Geospatial Center in providing WV image data. We thank two anonymous referees and Tom Neumann for their comments.

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

| Lake | dV, $km^3$ | local melt, $km^3 yr^{-1}$ | total melt, $km^3 yr^{-1}$ | $T_{local}$, yr | $T_{total}$, yr |
|---|---|---|---|---|---|
| $Thw_{70}$ | 0.87 | 0.034 | 0.07 | 25 | 13 |
| $Thw_{124}$ | 3.7 | 0.045 | 0.17 | 83 | 22 |
| $Thw_{142}$ | 0.54 | 0.014 | 0.12 | 39 | 4.7 |
| $Thw_{170}$ | 0.49 | 0.0076 | 0.044 | 64 | 11 |

Table 1. Discharge estimate for each lake, as well as the local (within-basin) and total (within-basin + upstream) melt supplies to each lake, and the time required for local and total melt supplies to refill the water discharged during the lake drainage ($T_{local}$ and $T_{total}$, respectively).

| | Values considered | Value chosen |
|---|---|---|
| $E\left(\dfrac{\partial^2 z_0}{\partial x^2}\right)$ | $[1.5\times10^{-7} m^{-2} \ldots 1.5\times10^{-5} m^{-2}]$ | $3\times10^{-7} m^{-2}$ |
| $E\left(\dfrac{\partial^3 \delta z}{\partial x^2 \partial t}\right)$ | $[1.5\times10^{-8} m^{-2} yr^{-1} \ldots 1.5\times10^{-6} m^{-2} yr^{-1}]$ | $6\times10^{-8} m^{-2} yr^{-1}$ |
| $E\left(\dfrac{\partial^2 \delta z}{\partial t^2}\right)$ | $[0.01\ m\ yr^{-2} \ldots 10\ m\ yr^{-2}]$ | $1\ m\ yr^{-2}$ |

Table 2. Expected elevation statistics values used to choose weighting parameters in (20). Here E() indicates the expected value of a quantity.

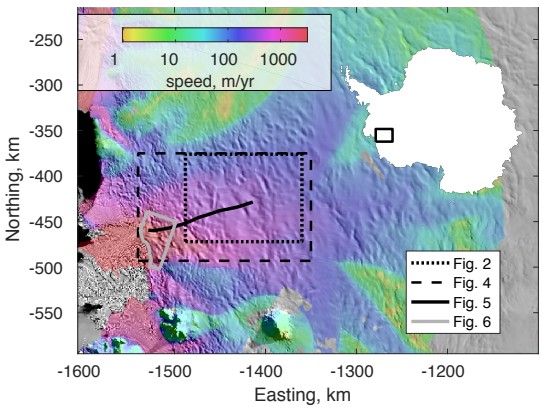

**Figure 1. Location map, showing an image mosaic (Scambos and others, 2007) and surface speed (Rignot and others, 2011), and locations for Figures 2-6. Northing and Easting are in a polar stereographic projection with a standard latitude of -71 S.**

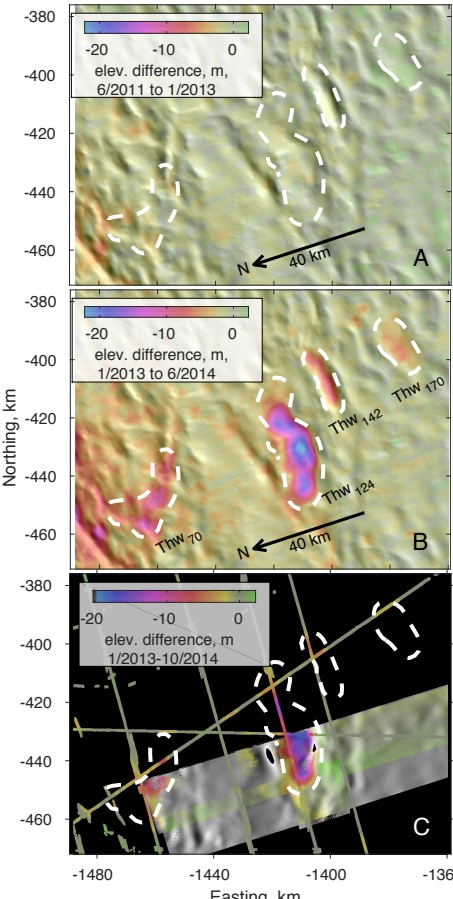

**Figure 2. Elevation and elevation change for our study area on Thwaites Glacier. The background shading in A and B is derived from the surface slope of the June-2011 reference DEM derived from CryoSat altimetry. The region mapped corresponds to the dotted box in Figure 1. A: Elevation changes derived from CryoSat altimetry between June 2011 and January 2013. B: Elevation changes derived from CryoSat altimetry between January 2013 and June 2014. C. Elevation change recovered from WV DEM and IceBridge laser altimetry differencing, on a background showing the slope of the WV DEMs. Dashed outlines show feature boundaries.**

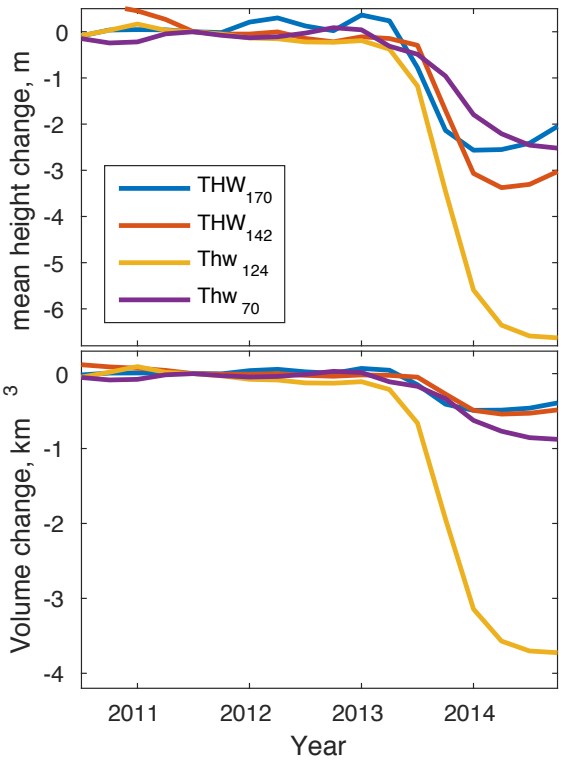

**Figure 3. Top: Mean elevation change relative to 1 June, 2011 within the digitized outlines from Figure 2, corrected for elevation change outside the outlines. Bottom: Calculated volume change within the outlines.**

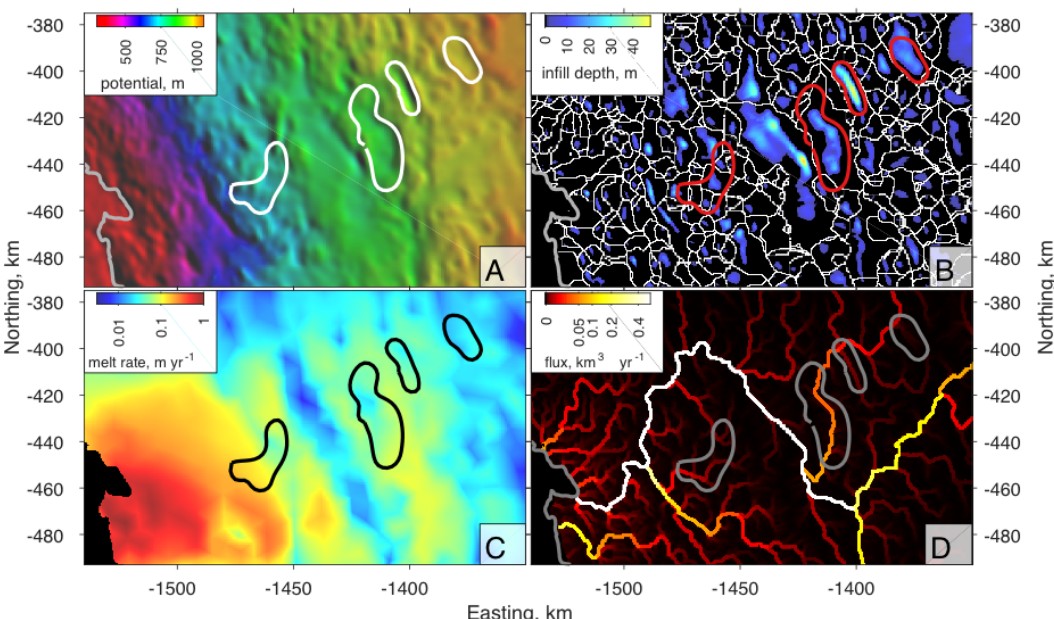

Figure 4. Quantities related to meltwater production and routing, showing that the lakes coincide with a prominent drainage path that connects to the grounding line. The mapped area corresponds to the dashed box in Figure 1. A: Hydropotential map derived from the June, 2011 surface elevation map and our basal topography map. B: Merged basins derived from the hydropotential map, and the water-filling depth required to eliminate local water sinks. C. Melt-rate estimate derived from estimated basal shear stress and sliding speed (Joughin et al., 2009). D. Water-flux magnitude derived from the basal-melt map and the filled hydropotential map. The grounding-line position (Rignot et al., 2011a; Rignot et al., 2011b) is shown in grey in A, B, and D.

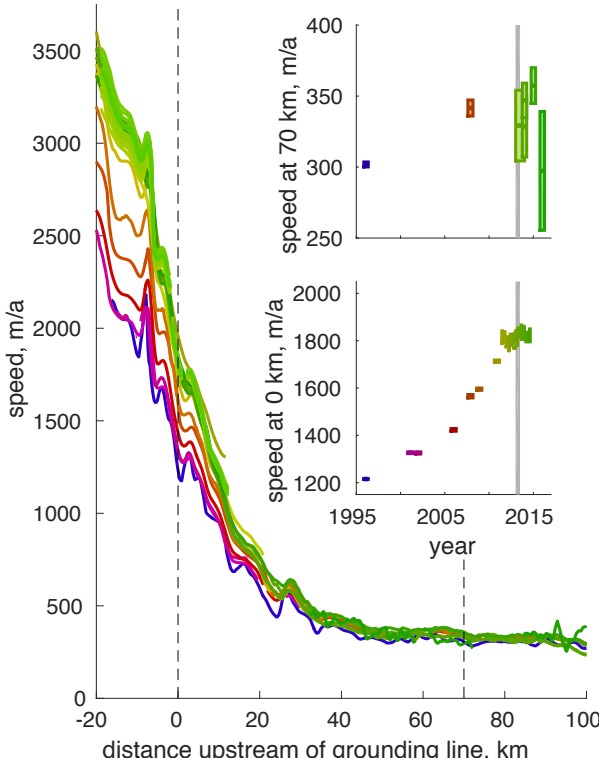

**Figure 5. Glacier surface speeds along a profile running from the grounding line through the draw-down features, plotted as a function of distance upstream of the grounding line. Insets show speeds as a function of time at the downstream end of Thw$_{70}$, and at the grounding line (dashed vertical lines). Lines are colour coded by time, with the complete range of colours shown in the grounding-line (0-km) speed-vs-time plot. The grey bar in the inset indicates the time of the lake drainage. Speed changes at both the grounding line and the downstream end of the lake system (km 70) are small compared to the long-term speedup of the glacier.**

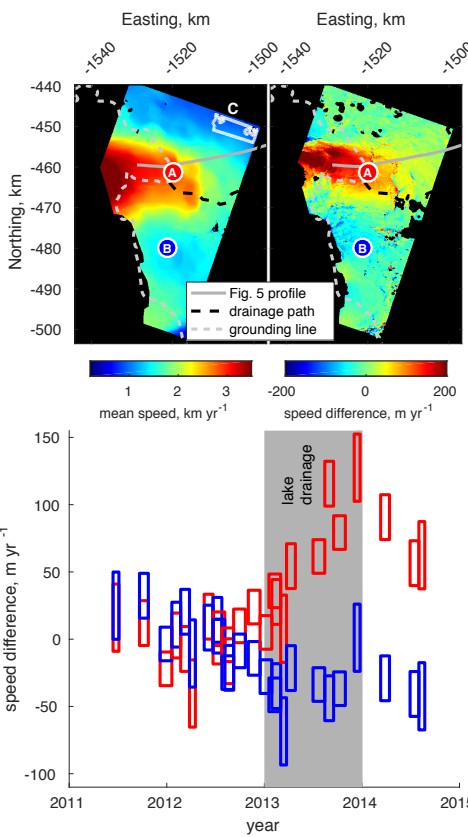

**Figure 6. Detail showing speed change at the grounding line, based on TSX and TDX SAR velocities. Upper left: the mean speed between mid 2011 and late 2012. Upper right: Speed difference between the August 27, 2013 speed map, and the 2011-12 mean speed. Bottom: Speed change relative to the 2011-12 mean for 'A' and 'B'. The area with the largest speed change is close to the outlet of the largest drainage path (Figure 4), and the strongest acceleration coincided roughly with the lakes' drainage.**