# Peer review of "Connected subglacial lake drainage beneath Thwaites Glacier, West Antarctica"

_The Cryosphere, 2016_

## Referee Comment (RC1) · Anonymous Referee #1 · 5 Sep 2016

General comments

The paper describes observations that indicate the existence of connected subglacial lakes beneath Thwaites Glacier. The paper presents the first evidence for subglacial lakes in this region well, in a balanced way in the context of existing literature. As a result I support the publishing of the manuscript with only minor changes.

One question that I had, however, was what the return looks like in the radar data from the AGASEA/Icebridge surveys in this area? I assume there are survey lines that cross these lakes? Is there any evidence supporting the existence of the lakes? Given the timescales of filling and draining in the paper the lake should have been existent at the time of the AGASEA survey?

Secondly, I find that the conclusion makes a number of unqualified statements about

subglacial hydrology in general from the conclusions from the Thwaites Glacier observations. You have observations from one system over one period of time which you then use to make sweeping statements about the importance of subglacial hydrology in general. I suggest you modify the conclusion to qualify some of these statements. In particular, page 9 lines 37- page 10 line 2.

Specific comments

Page 1, line 22: Add some references for the AGASEA and IceBridge campaigns

page 3, line 18: remove the second "was generated" from this sentence

page 4, end of line 7: remove "a"

page 4, line 10: add high <to> low

page 5, line 14: remove "we derived"

page 8, line 5: change "lakes drainage" to "lake drainage"

page 8, line 14: change "its" to "it is"

---

## Referee Comment (RC2) · Anonymous Referee #2 · 30 Sep 2016

**1  General Comments**

The authors demonstrate for the first time the existence of active lakes in the fast flowing Thwaites Glacier catchment akin to those identified through InSAR and repeat laser altimetry in the Siple Coast and East Antarctica. Most persuasive is the signal seen in the repeat WorldView DTM data, which unfortunately has limited coverage. If the authors address concerns related to their integration of two Cryosat products (detailed below) this paper will provide valuable observations on the distribution of these features. In addition, any potential signal seen from IceBridge flights over these targets should be addressed. The comparison with grounding line velocities is informative.

There are problems with the discussion and conclusions. There are gaps in the litera-

ture addressed, and some apparent confusion about literature that is cited. The paper is strongest on the observational side; much of the hydraulic routing is not novel, and doesn't matter that much to their conclusions - I would suggest focusing the paper more on the observations and less on the routing and recurrence time arguments. The statement in the conclusion that subglacial hydrology is not important is to focus to much on the transient response using the Byrd Glacier paradigm, and to make assumptions about what is fundamentally organizing the basal shear stress on the bed.

Figures could do with a work over (detailed below) for clarity, and much of the terminology for datasets needs to be rendered consistent. **While the primary observation is clearly worthy of publication in the Cryosphere, I recommend major revisions.**

**2  Specific Comments**

**page 1: line 29** Schroeder et al., 2013 (doi:10.1073/pnas.1302828110) and 2015 (doi:10.1109/LGRS.2014.2337878) were explicit that the observed basal hydrology was highly collimated large aspect ratio canals, a little bit different from "small pockets". Notably, as can be seen in figure 2B of Schroeder et al., 2013, and from Young et al., 2015 (doi:10.1098/rsta.2014.0297) the region of the proposed lakes lies within the region of the anisotropic water system. The geometries inferred from the 2005 radar in Schroeder et al., 2015 are difficult to reconcile with the amount of storage inferred by the 2014 observations. The authors might want to place this lake observation in context of these other papers.

**2: 4** It appears that there are two IceBridge ICESat reflight lines (OIB 20111112 and 20141122) that crossed these features with ATM data spanning the interval in question **- the authors should either perform that straightforward dtdz comparison or explain why it is invalid.**

**2: 21-38** A big deal is made of the combined use of the POCA and swath products, but

there is little representation of where POCA and swath products are used; in particular for where these products are with respect to the lakes. **I suggest that the authors add a figure for the 2011 DEM showing where POCA returns and swath points are wrt the lake outlines.**

The WorldView product validates to the dzdt result, however it seems the (apparently unbiased) POCA will cluster on the highs, and swath (with significant inter-season biases) should fill the topographic lows - exactly where the majority of the dHDt is observed. Note that the simulated image in Figure S2 will primarily respond to the highs that will be well mapped by POCA, and not have much as signal for the local, flat lows mapped by swath. **On line 34, the source of the DEM that the ambiguous swath measurement is compared with should be explicitly stated.**

Grima et al., 2014 (doi:10.1002/2014GL061635) point out that this exact area of Thwaites Glacier has considerable variability in firn density (notably one detected at radio frequencies due to variations in dialectic contrast) that is related to surface slope. As the steepest surface slopes (and higher density firn) bound the features, its seems plausible that low density firn preferentially fills the lake features. **The authors should present a case that either time varying penetration of low density firn or actual densification of low density firn does not represent part of the lower signature.**

**3:22** Provide a citation for the laser altimetry datasets

**6:33 The Bedmap2 derived flow routing should be shown in supplementary materials, in addition to the comparison bed and hydraulic maps.**

**7:7-8** *"Before this acceleration, this area was slowing at about 50 m yr$^{-2}$, and after the start of 2014 it returned to this slowing rate."* The sentence is difficult to follow because the reader is tasked with keeping track of four demonstratives. **Reword for clarity by explicitly stating what "this", "this", "it", and "this" mean.**

**7:14** This section is a completely incorrect representation of the Siegert et al 2014

paper. Seigert et al., 2014 based on radar observational concurred with the uncited Sergienko and Hulbe, 2011, (doi:10.3189/172756411797252176) that fast flowing ice streams subglacial water would cling on the lee side of subglacial topography, rather than forming a classic subglacial lake - a result that is supported by this work (the inferred lakes are all hanging off of bedrock ridges, rather than siting in the middle of bedrock basins). **Section should be rewritten after a more careful rereading of Seigert et al 2014 and Sergienko and Hulbe, 2011.**

**7:17** A quantitative value for the volume of subglacial material is mentioned for the first time here, but the authors have not been clear about how the subglacial volume has been calculated. We are left to assume that the authors have equated surface elevation change with subglacial volume change. If that's true, state it explicitly. Sergienko et al. 2007 (doi.org/10.1029/2007GL031775) argue that the surface volume change corresponding to a subglacial lake drainage event should not be conflated with the volume of subglacial water drainage, although it may be admissible if there is not change in velocity. **Explicitly state how surface measurements have been used to estimate subglacial water volumes, and provide appropriate justification.** Also remove the hyphen from "4-km$^3$ volume".

**8:12** *"With this model, and upstream lake could overflow into a downstream lake, which would subsequently cause it to overflow, which would trigger the next event."* The process described here and the methods used to observe the process are quite similar to Flament et al. 2014 (doi.org/10.5194/tc-8-673-2014), yet there is no mention of the Flament et al. paper anywhere in this manuscript. **Cite Flament.**

**8:30** The steady state method routing of Schroder et al., 2014 (doi:10.1073/pnas.1405184111), as stated in that paper, only was applied to regions where radar reflectivity as of 2005 indicated that hydrostatic canals with smooth interfaces dominated the bed echo return. In addition, its important to say in this context that transient lakes such as these have not been shown to have a strong enhanced radar reflectivity signature - while the geothermal flux method of Schroder

et al., 2014 is relying on the spatial variability of the "background" reflectivity signature of the hydrostatic canals, as they cover more or less of the bed.

**9:23** The Conclusions section begins by mentioning a value of $>3.5$ km$^3$ for subglacial water volume, although this value did not appear anywhere in the Results section. It is unclear whether $>3.5$ km$^3$ refers to the 4 km$^3$ mentioned on page 7, line 17. Do these different values represent the same physical quantity? Why don't they agree? **Clarify.**

**9:37** The logic that the subglacial water system does not matter much because of the lack of response to the individual drainage event is flawed. As the authors point out, (and is pointed out in Sergienko et al., 2014), much of the basal drag in this system is restricted to distinct bands, which control the stress state and flow of the glacier. The conclusion of Schroder et al., 2013 was that in these high drag zones, more water would not affect bed coupling (even if it was episodic). However, much of the ice flow between these bands is currently over sliding bed with distributed water systems. The argument of Schroder et al., 2013 is that it is the transformation of these distributed water systems into channelized flow (like the current high drag bands) that would change the stress state of the entire system.

**3 Technical Corrections**

1:17 TWG is not defined and is not used anywhere else in the manuscript.

1:21 and throughout the manuscript "Thwaites glacier" should be "Thwaites Glacier".

2:21 and throughout the manuscript "Cryosat-2" should be "CryoSat-2".

2:30 comma needed; change "$-2\pi 0,$ and $2\pi$" to "$-2\pi, 0,$ and $2\pi$".

3:10 "AMES" should be "Ames".

3:18 Fix "We generated a bed DEM was generated based on..."

3:19 and elsewhere "BEDMAP-2" should be "Bedmap2".

3:21 MCoRDS is miscapitalized and misspelled.

4:29 "LANDSAT" should be "Landsat".

4:29 TSX is defined but not consistently used later.

4:32 "Landsat-8" should be "Landsat 8".

5:31 "Worldview-2" should be "WorldView-2".

5:32 Inconsistent lake naming: "$THW_{124}$ and $Thw_{70}$" should be "$Thw_{124}$ and $Thw_{70}$".

6:3 Two issues here: Previous sub-figures have been identified with capital letters, but here "Figure 3a" is identified with a lowercase "a". Inspection of Figure 3 reveals no panels labeled "a" or "A".

6:33 "Bedmap-2" should be "Bedmap2".

6:34 and throughout the manuscript Capitalization of the word "figure" is not consistent. On this page we have "figure 4C" and "figure 5", but elsewhere in the manuscript (e.g., page 2 line 7) we see the more common convention of capitalizing "Figure". Whichever capitalization is chosen, it should be consistent and capitalization of the word "Table" (e.g., page 6, line 37) should match.

7:37 Change "there is uncertainty our" to "there is uncertainty in our".

8:4 A sentence begins "Despite these limitations..." What limitations?

8:5 Change "the lakes drainages" to "the lake drainages".

8:5 Change "where some of deepest closed basins" to "where some of the deepest closed basins".

8:6 and elsewhere The word that previously appeared in the manuscript as "figure" or "Figure" now appears as "Fig" without a period and occurs later on line 10 as "Fig."

with a period. Be consistent.

8:10 Figure 3d is referenced, although no such figure exists.

8:14 Change "its inconsistent" to "it's inconsistent" or "it is inconsistent".

8:14 The word "draining" should be "drained", but for readability consider changing "...which suggest, although not definitively, $Thw_{124}$ drained first." to "which suggests $Thw_{124}$ likely drained first."

8:15 It is not clear what process the word "this" refers to in the phrase "this should not trigger the other lakes" .

8:27 Change "by substantially short paths than shown" to "by substantially shorter paths than shown".

8:41 Remove the period after (Joughin et al., 2009).

9:26 The primary quantitative results of this paper have changed yet again, as subglacial water volume is now listed as $3 \text{ km}^3 - -25\%$ less than its original value.

10:7 The acronym stands for "Ice, Cloud, and land Elevation Satellite".

13:7 This is the second equation numbered 19. Be sure to fix the caption of Table 2 accordingly.

13:37 "terrasar-X" should be "TerraSAR-X".

**Table 1** Headings $T_{local}$ and $T_{total}$ should be explicitly defined in the caption.

**Table 2** The letter E should be explicitly defined in the caption.

**Figures** In general, the figure captions don't contain enough information to describe the figures on their own. This is a problem for people who like to skim the figures before reading the paper.

**Figure 2** Mention region is the box in fig 1? Is elevation shown as the shading? If so,

[Figure]

which elevation was used? Mention that A and B are cryosat, then give dates Maybe add labels to the 4 lakes, since they're used in Fig 3. Fig 2c There is a green streak that appears to be a correlated error

**Figure 2 caption** "Worldview" should be "WorldView-2" or use the acronym that was introduced in the main text.

**Figure 3** Mention that outlines are from Fig 2. "mean elevation change" with respect to what?

**Figure 4** This isn't quite the same outline as shown in Figure 1 for Figure 2. Please provide a context map. What GL are you plotting here? Mention how the melt-rate was derived. "melt rate from basal shear"

**Figure 4 caption** Rather than simply, "C. Melt-rate estimate." remind readers how melt rates were estimated, or what dataset is plotted. An added suggestion to improve this figure and others: it seems the subplot titles have been left out of the figure itself and have been moved to the caption, where they displace meaningful information and task the reader with keeping track of which subplot is which. Figure captions provide an opportunity to describe processes, to give the reader clues about what we should be seeing, to give insight and understanding. Instead, in this figure caption and in others all we are given is a list of sentence fragments that would be more appreciated as subplot titles.

**Figure 5** This suggestion may end up in a too-cluttered figure, but it would be helpful to know which platforms were used to obtain the different velocity measurements. I'd like to have seen dotted lines (or grey bars) for the lake locations Mention that grey bar in inset is the drainage event.

**Figure 6** Include AB labels on the right image As mentioned before, I'm worried about region C's location relative to the drainage pathways and where you'd expect velocities to be changing.

**Figure 6 caption** "Terrasar-X" should be "TerraSAR-X". Include AB labels on the right image As mentioned before, I'm worried about region C's location relative to the drainage pathways and where you'd expect velocities to be changing.

**In the supplemental data** `bed_DEM.tif` was identical to `surface_DEM.tif`

`Thw_lakes_outline.gmt` had severe parsing problems in gdal with leading spaces and the additional commented lines - a simple ASCII table would be preferable.

---

## Editor Comment (EC1) · J. L. Bamber (Editor) · 3 Oct 2016

Dear author, you have received two constructive reviews that see promise in the results you present and make a number of useful suggestions for improving the manuscript. I encourage you to submit a revised m/s highlighting how you have addressed the referees comments and concerns.

---

## Author Comment (AC1) · 5 Nov 2016

Response to reviewer 1.

We thank the referee for the encouraging assessment of our paper, and respond to his or her comments below. Referee 1's comments are prefixed by "R1:", our responses by "Au:"

R1: One question that I had, however, was what the return looks like in the radar data from the AGASEA/Icebridge surveys in this area? I assume there are survey lines that cross these lakes? Is there any evidence supporting the existence of the lakes? Given the timescales of filling and draining in the paper the lake should have been existent at the time of the AGASEA survey? Au: The AGASEA radargrams are not, to our knowledge, publically available, and our cursory examination of the IceBridge

radargrams did not show anything remarkable close to the lakes. This is not unusual: Authors who have looked at radargrams over active subglacial lakes (e.g Siegert et al., 2014) have often not seen strong radar signatures. This is likely because the roofs of lakes in fast-flowing areas retain the imprint of the last bed topography the ice encountered before it moved over the lake, so the bottom of the ice over the lake is not as smooth as it would be if the ice were moving slowly and the ice sole had time to flatten. This discussion is outside the scope of our paper however, so we do not include it.

R1: Secondly, I find that the conclusion makes a number of unqualified statements about subglacial hydrology in general from the conclusions from the Thwaites Glacier observations. You have observations from one system over one period of time which you then use to make sweeping statements about the importance of subglacial hydrology in general. I suggest you modify the conclusion to qualify some of these statements. In particular, page 9 lines 37- page 10 line 2. Au: We have narrowed these conclusions in response to both reviewers'comments, and now say: While our data suggest water is routed in ways not presently accounted for in most ice sheet models, it also indicates that changes of this type in the basal hydrological system may not matter much. The basal water system is able to sequester large volumes of water over years which it then releases rapidly with little or no apparent change in glacier speed. This insensitivity suggests that the details of the basal hydrological system may not be the most important feature of the ice sheet for models to capture, especially now that data assimilation techniques allow us to infer the dynamic properties of the bed (e.g., the coefficients in a sliding law) directly (Joughin et al., 2010; Morlighem et al., 2010). At least at the decadal scale, fixed bed parameters can reasonably reproduce observed behaviour (Joughin et al., 2010; Joughin et al., 2014), despite large increases in water-layer thickness that accompany a speedup and lake drainages. The lack of sensitivity is probably related to the patchy structure of basal drag beneath TWG, and the limited time over which lake drainages supply water. As previous studies have noted (Joughin et al., 2009; Sergienko and Hindmarsh, 2013) much of the drag restraining the ice flow

is concentrated in small patches or bands, and if changes in water pressure reduce the drag in the low-drag areas between these patches, the speed of the glacier is unlikely to change significantly. Further, a short-duration drainage, even of a large volume of water, cannot cause a large change the long-term average discharge of a fast-flowing glacier like THW. With only a few examples of changes in water availability to Antarctic glaciers documented, data are too sparse at present to say definitively whether an evolving hydrological system is an essential part of a predictive ice sheet model. Nevertheless, the data that do exist suggest that such sensitivity to hydrological evolution may be small.

R1: Page 1, line 22: Add some references for the AGASEA and IceBridge campaigns. Au: We did not have references to the AGASEA and IceBridge campaigns in the original draft of the paper, but have added them in section 2 (the data description).

R1: page 3, line 18: remove the second "was generated" from this sentence Au: Fixed.

R1: page 4, end of line 7: remove "a" Au: We changed "Bed DEMs" to "Bed DEM," which fixed the problem.

R1: page 4, line 10: add high <to> low Au: Fixed

R1: page 5, line 14: remove "we derived" Au: Fixed.

R1: page 8, line 5: change "lakes drainage" to "lake drainage" Au: Fixed.

R1: page 8, line 14: change "its" to "it is" Au: Fixed.

---

## Author Comment (AC2) · 6 Nov 2016

Response to reviewer 2.

We thank reviewer 2 for the extensive comments provided on our manuscript, and hope that we have addressed them adequately below.

R2:page 1: line 29 Schroeder et al., 2013 (doi:10.1073/pnas.1302828110) and 2015 (doi:10.1109/LGRS.2014.2337878) were explicit that the observed basal hydrology was highly collimated large aspect ratio canals, a little bit different from "small pockets". Notably, as can be seen in figure 2B of Schroeder et al., 2013, and from Young et al., 2015 (doi:10.1098/rsta.2014.0297) the region of the proposed lakes lies within the region of the anisotropic water system. The geometries inferred from the 2005 radar in Schroeder et al., 2015 are difficult to reconcile with the amount of storage inferred

[Figure]

by the 2014 observations. The authors might want to place this lake observation in context of these other papers.

Au: We now comment on this in 4.1: Over much of the area around the lakes, characteristics of radar returns from the bed have led researchers to infer the presence of a basal drainage system comprised of elongated channels running parallel to ice flow (Schroeder et al., 2015; Schroeder et al., 2013). Such a system of elongated canals could prevent the accumulation of large volumes of water if it were broadly connected, so our results suggest that if a large-scale canal system is present around the lakes, there may be gaps in its spatial connections, or it may not have sufficient conductivity to prevent large lakes from accumulating

R2: 2: 4 It appears that there are two IceBridge ICESat reflight lines (OIB 20111112 and 20141122) that crossed these features with ATM data spanning the interval in ques- tion - the authors should either perform that straightforward dtdz comparison or explain why it is invalid.

Au: We now include OIB elevation differences (2.3) and compare the results to the WV DEM differences (3.1, and figure 2). The results are fairly similar between the WV and OIB results, and the OIB elevation differences show signals similar to the Cryosat differences for the two upper lakes.

R2: 21-38 A big deal is made of the combined use of the POCA and swath products, but there is little representation of where POCA and swath products are used; in particular for where these products are with respect to the lakes. I suggest that the authors add a figure for the 2011 DEM showing where POCA returns and swath points are wrt the lake outlines. The WorldView product validates to the dzdt result, however it seems the (apparently unbiased) POCA will cluster on the highs, and swath (with significant inter-season biases) should fill the topographic lows - exactly where the majority of the dHDt is observed.

Au: We now include a map of the point density for the two products in the supplemental

material, and include a comment on the coverage in section 3.1: A map of the density of elevation measurements remaining after our iterative editing process (Figure S3) shows that while POCA measurements tended to cluster on local highs on the surface while swath measurements are more broadly distributed, points from each of the two sets of measurements contribute to elevation estimates within the outlines. This shows that both types of data contribute to the measured elevation changes, and that the elevation differences are not solely due to bias changes in the swath-processed data.

R2: Note that the simulated image in Figure S2 will primarily respond to the highs that will be well mapped by POCA, and not have much as signal for the local, flat lows mapped by swath.

Au: It is probably true that the areas covered most densely by the POCA data have the largest slopes, but this does not necessarily imply that the simulated image in figure S2 is determined only by the POCA data. It is also significant that the elevation-fitting strategy produced a smooth surface in areas where the ice sheet is smooth; a strategy that did not work as well might have produced a uniformly rough surface, or produced features in areas that are in fact flat.

R2: On line 34, the source of the DEM that the ambiguous swath measurement is compared with should be explicitly stated.

Au: We now identify the DEM as: based on mosaicked WV DEMs (Shean et al, 2016) and IceBridge altimetry.

R2: Grima et al., 2014 (doi:10.1002/2014GL061635) point out that this exact area of Thwaites Glacier has considerable variability in firn density (notably one detected at radio frequencies due to variations in dialectic contrast) that is related to surface slope. As the steepest surface slopes (and higher density firn) bound the features, its seems plausible that low density firn preferentially fills the lake features. The authors should present a case that either time varying penetration of low density firn or actual densification of low density firn does not represent part of the lower signature.

Au: We now treat these possible signals explicitly: Near-surface density can vary in time (Ligtenberg et al., 2011), and these variations are can cause both real surface-elevation changes and apparent surface-elevation changes due to changes in the penetration of radar altimeters' energy into the firn (Ligtenberg et al., 2012). At the same time, firn density likely varies on short spatial scales on Thwaites glacier, driven in part by surface slope variations (Grima et al., 2014). These two effects together might lead to apparent surface-elevation changes in CryoSat data, on the spatial scale of the changes observed here. The close agreement between the surface-elevation changes measured by CryoSat, laser altimetry, and photogrammetry in the areas where the three overlap suggest strongly that the CryoSat changes reflect real changes in the surface height, and not temporal changes in subsurface penetration of radar energy. Given that the surface elevation likely changed by several meters, it seems unlikely that changes in firn density alone could have produced these changes. The total range of estimated firn-air content change for this area between 1979 and 2012 is less than 1 m (Ligtenberg et al., 2014), much smaller than the 4-20 m changes observed here.

R2:3:22 Provide a citation for the laser altimetry datasets

Au: Done.

R2:6:33 The Bedmap2 derived flow routing should be shown in supplementary materials, in addition to the comparison bed and hydraulic maps.

Au: We will include this in our revised submission.

R2:7:7-8 "Before this acceleration, this area was slowing at about start of 2014 it returned to this slowing rate." The sentence is difficult to follow because the reader is tasked with keeping track of four demonstratives. Reword for clarity by explicitly stating what "this", "this", "it", and "this" mean.

Au: We reworded this sentence a bit:

These maps show that a small area, about 15x20 km in extent, on the east side of the

glacier, accelerated by about 100 m yr-1 over the course of the 2013 calendar year, then slowed by about half as much over the course of 2014.

R2: 7:14 This section is a completely incorrect representation of the Siegert et al 2014 paper. Seigert et al., 2014 based on radar observational concurred with the uncited Sergienko and Hulbe, 2011, (doi:10.3189/172756411797252176) that fast flowing ice streams subglacial water would cling on the lee side of subglacial topography, rather than forming a classic subglacial lake - a result that is supported by this work (the inferred lakes are all hanging off of bedrock ridges, rather than siting in the middle of bedrock basins). Section should be rewritten after a more careful rereading of Seigert et al 2014 and Sergienko and Hulbe, 2011.

Au: We regret misattributing the idea presented to the Siegert et. al. paper. For the sake of simplicity, we have removed the remark about the movement of water or till. We have added material to 4.1 that addresses some of the ideas in the Sergienko and Hulbe and the Siegert et al papers:

Previous studies (Bindschadler and Choi, 2007; Siegert et al., 2014) have identified locations such as these as likely to trap water, and have shown that even on smooth beds, surface topography generated by local variations in basal traction can produce hydropotential basins that trap water (Sergienko and Hulbe, 2011). Otherwise, it is not clear that our paper is at all in conflict with the Sergienko and Hulbe paper or the Siegert et al paper. Both used the Shreve potential to estimate were lakes might be. Sergienko and Hulbe explored a different way in which sticky spots might give rise to surface topography, that would then modify the Shreve potential, but the way we mapped the hydropotential is agnostic as to whether the surface topography was generated by basal traction variations or by bumps on the bed. It is very likely that both play a role, but it does not affect our analysis.

R2: 7:17 A quantitative value for the volume of subglacial material is mentioned for the first time here, but the authors have not been clear about how the subglacial volume has been calculated. We are left to assume that the authors have equated surface elevation change with subglacial volume change. If that's true, state it explicitly. Sergienko et al. 2007 (doi.org/10.1029/2007GL031775) argue that the surface volume change corresponding to a subglacial lake drainage event should not be conflated with the volume of subglacial water drainage, although it may be admissible if there is not change in velocity. Explicitly state how surface measurements have been used to estimate subglacial water volumes, and provide appropriate justification. Also remove the hyphen from "4-km3 volume".

Au: We now include a discussion of this mechanism: Detailed modelling of the surface changes associated with changes in basal topography (Gudmundsson, 2003; Sergienko et al., 2007) show that in fast-flowing environments, ice flow changes in response to perturbations in the surface shape can reduce the amplitude of surface elevation change in response to changes at the bed. Specifically, a lake that drains at the bed will produce a surface depression, but ice flowing into the depression will quickly reduce its depth. The net volume of the ice sheet must be conserved, so that the volume of the depression at the surface must equal volume drained at the bed, but as ice flow refills the lake depression, and the correction we make for regional uplift or drawdown could lead us to an overall underestimate of the lake volume change. This suggests that the volume of water displaced at the glacier bed during the lake drainages was larger than the volume of the changes at the surface, and that our measurements represent a minimum estimate of the water movement. Lacking any technique for estimating the relationship between the two volumes, we proceed as if they were equal, but acknowledge that that there is uncertainty in this approximation. By contrast, changes in basal drag (i.e. the appearance or disappearance of sticky spots) can produce changes in surface topography, but these changes should appear as dipole-like patterns oriented in the along-flow direction, with no net volume change (Gudmundsson, 2003). We do not see evidence of this kind of pattern in our altimetry measurements.

R2. 8:12 "With this model, and upstream lake could overflow into a downstream lake, which would subsequently cause it to overflow, which would trigger the next event." The process described here and the methods used to observe the process are quite similar to Flament et al. 2014 (doi.org/10.5194/tc-8-673-2014), yet there is no mention of the Flament et al. paper anywhere in this manuscript. Cite Flament

AU: We now cite Dr. Flament's paper in sections 3.1 and 4.1.

R2 8:30 The steady state method routing of Schroder et al., 2014 (doi:10.1073/pnas.1405184111), as stated in that paper, only was applied to regions where radar reflectivity as of 2005 indicated that hydrostatic canals with smooth interfaces dominated the bed echo return. In addition, its important to say in this context that transient lakes such as these have not been shown to have a strong enhanced radar reflectivity signature - while the geothermal flux method of Schroder et al., 2014 is relying on the spatial variability of the "background" reflectivity signature of the hydrostatic canals, as they cover more or less of the bed.

Au: The model in Schroeder et al, 2014 covers the area of Thwaites glacier south of 76 S, which includes all of the lakes considered here, except for the downstream part of Thw70. In that our comment is about how the Schroeder paper uses the assumption of steady-state flux, and not about the reflectance of the bed per se, we are not sure how to address the reviewer's comment here. We feel that the radar reflectivity is beyond the scope of this paper.

R2 9:23 The Conclusions section begins by mentioning a value of >3.5 km3 for subglacial water volume, although this value did not appear anywhere in the Results section. It is unclear whether >3.5 km3 refers to the 4 km3 mentioned on page 7, line 17. Do these different values represent the same physical quantity? Why don't they agree? Clarify

Au: We have revised the numbers on pages 7 and 9 to agree with the values in table 1. Thanks for recognizing the inconsistency.

R2 9:37 The logic that the subglacial water system does not matter much because of the lack of response to the individual drainage event is flawed. As the authors point out, (and is pointed out in Sergienko et al., 2014), much of the basal drag in this system is restricted to distinct bands, which control the stress state and flow of the glacier. The conclusion of Schroder et al., 2013 was that in these high drag zones, more water would not affect bed coupling (even if it was episodic). However, much of the ice flow between these bands is currently over sliding bed with distributed water systems. The argument of Schroder et al., 2013 is that it is the transformation of these distributed water systems into channelized flow (like the current high drag bands) that would change the stress state of the entire system.

Au: We have narrowed our conclusion, and included references to these studies. We now discuss the Schroeder paper in the discussion section:

Our results are largely in agreement with the hypothesis that water in the lower part of Thwaites Glacier can travel through channels (Schroeder et al., 2013), but the pre-drainage retention of water suggests that the channels are at most intermittently active. If the upstream lakes were briefly connected by a low-pressure channel, the lack of substantial glacier slowdown after the end of the subglacial flood suggests that the induced transition from a high-pressure distributed water system to a low-pressure channel was not permanent, or at least that it did not produce a substantial change in basal traction on the glacier.

We also restrict our suggestion about the importance of the basal water system conditions like those found on TWG:

While our data suggest water is routed in ways not presently accounted for in most ice sheet models, it also indicates that changes of this type in the basal hydrological system may not matter much. The basal water system is able to sequester large volumes of water over years which it then releases rapidly with little or no apparent change in glacier speed. This insensitivity suggests that the details of the basal hydrological system may not be the most important feature of the ice sheet for models to capture, especially now that data assimilation techniques allow us to infer the dynamic properties of the bed (e.g., the coefficients in a sliding law) directly (Joughin et al., 2010; Morlighem et al., 2010). At least at the decadal scale, fixed bed parameters can reasonably reproduce observed behaviour (Joughin et al., 2010; Joughin et al., 2014), despite large increases in water-layer thickness that accompany a speedup and lake drainages. The lack of sensitivity is probably related to the patchy structure of basal drag beneath TWG, and the limited time over which lake drainages supply water. As previous studies have noted (Joughin et al., 2009; 2013; Sergienko and Hindmarsh, 2013) much of the drag restraining the ice flow is concentrated in small patches or bands, and if changes in water pressure reduce the drag in the low-drag areas between these patches, the speed of the glacier is unlikely to change significantly. Further, a short-duration drainage, even of a large volume of water, cannot cause a large change in the long-term average discharge of a fast-flowing glacier like THW. With only a few examples of changes in water availability to Antarctic glaciers documented, data are too sparse at present to say definitively whether an evolving hydrological system is an essential part of a predictive ice sheet model.

Technical corrections:

R2: 1:17 TWG is not defined and is not used anywhere else in the manuscript.

AU: Corrected.

R2: 1:21 and throughout the manuscript "Thwaites glacier" should be "Thwaites Glacier".

AU: Corrected.

R2: 2:21 and throughout the manuscript "Cryosat-2" should be "CryoSat-2". AU: Corrected.

R2: 2:30 comma needed; change "ôĂĂĂ2_0; and 2_" to "ôĂĂĂ2_; 0; and 2_".

AU: Corrected.

R2: 3:10 "AMES" should be "Ames".

AU: Corrected.

R2 3:18 Fix "We generated a bed DEM was generated based on..."

AU: Corrected

R2: 3:19 and elsewhere "BEDMAP-2" should be "Bedmap2".

AU: Corrected

R2: 3:21 MCoRDS is miscapitalized and misspelled.

AU: Corrected

R2: 4:29 "LANDSAT" should be "Landsat".

AU: Corrected

R2:4:29 TSX is defined but not consistently used later.

AU: We now use TSX and TDX consistently throughout.

R2:4:32 "Landsat-8" should be "Landsat 8".

AU: The one occurrence of "Landsat-8" is a compound modifier on the word "imagery," so the hyphen is appropriate.

R2:5:31 "Worldview-2" should be "WorldView-2".

AU: Corrected

R2:5:32 Inconsistent lake naming: "THW124 and Thw70" should be "Thw124 and Thw70".

AU: Corrected

R2:6:3 Two issues here: Previous sub-figures have been identified with capital letters, but here "Figure 3a" is identified with a lowercase "a". Inspection of Figure 3 reveals no panels labeled "a" or "A".

Au: Fixed. The references should have been to figure 4 rather than figure 3.

R2:6:33 "Bedmap-2" should be "Bedmap2".

Au:Fixed

R2:6:34 and throughout the manuscript Capitalization of the word "figure" is not consistent. On this page we have "figure 4C" and "figure 5", but elsewhere in the manuscript (e.g., page 2 line 7) we see the more common convention of capitalizing "Figure". Whichever capitalization is chosen, it should be consistent and capitalization of the word "Table" (e.g., page 6, line 37) should match.

Au: Changed throughout to capital letters.

R2: 7:37 Change "there is uncertainty our" to "there is uncertainty in our".

Au: Fixed.

R2: 8:4 A sentence begins "Despite these limitations..." What limitations? Au: Changed to: "Despite evident limitations in our hydropotential maps' ability to predict water movement,"

R2:8:5 Change "the lakes drainages" to "the lake drainages"

Au: Fixed.

R2:8:5 Change "where some of deepest closed basins" to "where some of the deepest closed basins".

Au: Fixed, and reworded to be a bit less clumsy.

R2:8:6 and elsewhere The word that previously appeared in the manuscript as "figure" or "Figure" now appears as "Fig" without a period and occurs later on line 10 as "Fig."

with a period. Be consistent.

Au: Changed all to "Figure"

R2:8:10 Figure 3d is referenced, although no such figure exists.

Au: Changed to 4D

R2: 8:14 Change "its inconsistent" to "it's inconsistent" or "it is inconsistent".

Au: Changed to "it is"

R2:8:14 The word "draining" should be "drained", but for readability consider changing "...which suggest, although not definitively, Thw124 drained first." to "which suggests Thw124 likely drained first."

Au: Changed to match R2's suggestion.

R2:8:15 It is not clear what process the word "this" refers to in the phrase "this should not trigger the other lakes" .

Au: Changed and reworded, to:" In principle, the drainage of Thw124 should not trigger the drainage of the upstream lakes by the overflow mechanism, which would have to exceed their own potential barriers first."

R2: 8:27 Change "by substantially short paths than shown" to "by substantially shorter paths than shown".

Au: Fixed

R2:8:41 Remove the period after (Joughin et al., 2009).

Au: Fixed

R2: 9:26 The primary quantitative results of this paper have changed yet again, as subglacial water volume is now listed as 3 km3 ôĂĂĂ ôĂĂĂ25% less than its original value.

Au: Fixed

R2:10:7 The acronym stands for "Ice, Cloud, and land Elevation Satellite".

Au: Fixed.

R2:13:7 This is the second equation numbered 19. Be sure to fix the caption of Table 2 accordingly.

Au: Fixed.

R2: 13:37 "terrasar-X" should be "TerraSAR-X". Au: "terrasar-x" is no longer mentioned here.

R2: Table 1 Headings Tlocal and Ttotal should be explicitly defined in the caption.

Au: We added: "(Tlocal and Ttotal, respectively)" to the end of the sentence.

R2: Table 2 The letter E should be explicitly defined in the caption.

Au: Fixed.

R2: Figures In general, the figure captions don't contain enough information to describe the figures on their own. This is a problem for people who like to skim the figures before reading the paper.

Au: We have made changes to most of the figure captions in response to the comments below, and have moved material from the figure captions into the legends of figures.

R2:Figure 2 Mention region is the box in fig 1? Is elevation shown as the shading? If so, which elevation was used? Mention that A and B are cryosat, then give dates Maybe add labels to the 4 lakes, since they're used in Fig 3. Fig 2c. There is a green streak that appears to be a correlated error.

Au: We have added text to address these comments:

Figure 2. Elevation and elevation change for a study area around Thwaites Glacier.

The region mapped corresponds to the white box in figure 1. A: Elevation changes derived from CryoSat altimetry between June 2011 and January 2013. B: Elevation changes derived from CryoSat altimetry between January 2013 and June 2014. The background shading in A and B is derived from the surface slope of the June-2011 reference DEM derived from CryoSat altimetry. C. Elevation change recovered from WV DEM and IceBridge laser altimetry differencing, on a background showing the slope of the WV DEMs. Dashed outlines show feature boundaries.

R2: Figure 2 caption "Worldview" should be "WorldView-2" or use the acronym that was introduced in the main text.

Au: fixed.

R2: Figure 3 Mention that outlines are from Fig 2. "mean elevation change" with respect to what?

Au: We now note that the elevation differences are relative to 1 June 2011 and reference figure 2.

R2: Figure 4 This isn't quite the same outline as shown in Figure 1 for Figure 2. Please provide a context map. What GL are you plotting here? Mention how the melt-rate was derived. "melt rate from basal shear"

Au: We now include the location of figure 4 in figure 1, give a citation for the GL, and provide some words for how the melt rate was derived.

R2: Figure 5 This suggestion may end up in a too-cluttered figure, but it would be helpful to know which platforms were used to obtain the different velocity measurements. I'd like to have seen dotted lines (or grey bars) for the lake locations Mention that grey bar in inset is the drainage event.

Au: We acknowledge that it would have been nice to distinguish the different data sources, but did not find a way to indicate the different sources without cluttering further an already complicated figure. We now note in the caption that the position of

the grounding line and of Thw70 are marked by dashed lines, and that the grey bars indicate the lake drainage time.

R2: Figure 6 Include AB labels on the right image As mentioned before, I'm worried about region C's location relative to the drainage pathways and where you'd expect velocities to be changing.

Au: We added the labels, and added a line to represent the main drainage path from the hydropotential calculation. We hope that this resolves some of the worry.

R2: In the supplemental data bed_DEM.tif was identical to surface_DEM.tif

Au: We regret uploading the wrong file, and will fix this in the revised submission.

R2: Thw_lakes_outline.gmt had severe parsing problems in gdal with leading spaces and the additional commented lines - a simple ASCII table would be preferable.

Au: We have fixed the gmt format for this file, will also provide an ASCII table.

―――――――――――――――――――――